# Cystic fibrosis swine fail to secrete airway surface liquid in response to inhalation of pathogens

Xiaojie Luan[1], George Belev[2], Julian S. Tam[3], Santosh Jagadeeshan[1], Noman Hassan[1], Paula Gioino[1], Nikolay Grishchenko[1], Yanyun Huang[4], James L. Carmalt[5], Tanya Duke[6], Teela Jones[6], Bev Monson[7], Monique Burmester[7], Tomer Simovich [8], Orhan Yilmaz[1], Veronica A. Campanucci[1], Terry E. Machen[9], L. Dean Chapman [10] & Juan P. Ianowski[1]

Cystic fibrosis is caused by mutations in the gene encoding the cystic fibrosis transmembrane conductance regulator (CFTR) channel, which can result in chronic lung disease. The sequence of events leading to lung disease is not fully understood but recent data show that the critical pathogenic event is the loss of the ability to clear bacteria due to abnormal airway surface liquid secretion (ASL). However, whether the inhalation of bacteria triggers ASL secretion and whether this is abnormal in cystic fibrosis has never been tested. Here we show, using a novel synchrotron-based in vivo imaging technique, that wild-type pigs display both a basal and a Toll-like receptor-mediated ASL secretory response to the inhalation of cystic fibrosis relevant bacteria. Both mechanisms fail in CFTR$^{-/-}$ swine, suggesting that cystic fibrosis airways do not respond to inhaled pathogens, thus favoring infection and inflammation that may eventually lead to tissue remodeling and respiratory disease.

[1] Department of Physiology, University of Saskatchewan, Health Science Building, Room 2D01, 107 Wiggins Road, Saskatoon, SK, Canada S7N 5E5.
[2] Canadian Light Source Inc., 44 Innovation Boulevard, Saskatoon, SK, Canada S7N 2V3. [3] Department of Medicine, Division of Respirology, Critical Care, and Sleep Medicine, University of Saskatchewan, Royal University Hospital, 103 Hospital Drive, Saskatoon, SK, Canada S7N 0W8. [4] Prairie Diagnostic Services Inc., 52 Campus Drive, Saskatoon, SK, Canada S7N 5B4. [5] Department of Large Animal Clinical Sciences, Western College of Veterinary Medicine, University of Saskatchewan, 52 Campus Drive, Saskatoon, SK, Canada S7N 5B4. [6] Department of Small Animal Clinical Sciences, Western College of Veterinary Medicine, University of Saskatchewan, 52 Campus Drive, Saskatoon, SK, Canada S7N 5B4. [7] Animal Care Unit, Western College of Veterinary Medicine, University of Saskatchewan, 52 Campus Drive, Saskatoon, SK, Canada S7N 5B4. [8] Surface Science and Technology Group, School of Chemistry, The University of Melbourne, Parkville, VIC 3010, Australia. [9] Department of Molecular and Cell Biology, University of California, 231 LSA, Berkeley, CA 94720-3200, USA. [10] University of Saskatchewan, Department of Anatomy and Cell Biology, Health Science Building, Room 2D01, 107 Wiggins Road, Saskatoon, SK, Canada S7N 5E5. Correspondence and requests for materials should be addressed to J.P.I. (email: juan.ianowski@usask.ca)

C ystic fibrosis (CF) is an autosomal recessive genetic disorder caused by mutations in the *CFTR* gene. Although CF is a multisystemic disease, much of the morbidity and mortality arises from lung disease[1]. CF patients can develop chronic bacterial pulmonary infections and inflammation that may eventually cause end-stage lung disease. However, the sequence of events that result in lung disease from cystic fibrosis transmembrane conductance regulator (CFTR) mutations is not fully understood. Recent data show that the first outcome of CFTR mutation is an impaired ability to eliminate bacteria. Lungs from CF animal models (F508del and CFTR$^{-/-}$ pigs)[2, 3] fail to purge bacteria as effectively as wild-type littermates before the development of inflammation[3, 4]. These results suggest that impaired bacterial elimination is the pathogenic event initiating a cascade of inflammation and pathology in CF lungs[3].

The failure to clear bacteria seems to result from abnormal airway surface liquid (ASL) layer in the CF lung[5–8]. The ASL is a complex mixture of water, salts, gel-forming mucins, and antimicrobial compounds that contribute to trapping, inactivating, killing, and clearing pathogens through mucociliary clearance[9–11]. Airway samples from CF patients[7], 1-day old CFTR$^{-/-}$ piglets[6], newborn CFTR$^{-/-}$ ferrets[5], and CFTR$^{-/-}$ mice[8] fail to respond to stimulatory signals that normally elicit strong ASL secretion[12–15], suggesting that CF airway would fail to respond to inhaled insults[5–8]. It has therefore been proposed that CF airway may fail to secrete ASL in response to bacterial inhalation, thus reducing the amount of antimicrobial compounds and mucin available in the airway, compromising mucociliary clearance and facilitating infection and inflammation[4, 6–8, 15–17]. However, whether inhalation of bacteria triggers ASL secretion and whether this response is abnormal in CF lungs, has never been tested. We used CFTR$^{-/-}$ swine (gut-corrected CFTR$^{-/-}$, Fig. 1a, see "Methods" section) to study how CF airway respond to inhalation of bacteria in vivo, where all components of the regulatory mechanisms are intact, i.e., autonomic[12], sensory efferent[13], and immune signals[14, 15]. Our results show that there are a basal and a bacteria-triggered TLRs-related ASL secretory responses in the airway of wild-type swine. Both responses are missing in CFTR$^{-\backslash-}$ swine, which do not display basal secretion nor respond to inhalation of *Pseudomonas aeruginosa*. The results suggest the missing ASL secretory responses in CF airway leads to bacteria clearance failure, which further results in facilitating infection and inflammation.

## Result

**Synchrotron-based x-ray phase contrast imaging**. The main challenge in studying airway secretion in vivo is the lack of an imaging method with sufficient contrast and resolution to observe the thin ASL layer ($93 \pm 8\,\mu m$ thick, see Fig. 1) that is constantly cleared out of the airway through mucociliary clearance. We developed a synchrotron-based x-ray imaging technique to measure the ASL layer in vivo. The technique exploits the large difference in x-ray refractive index between the air and the ASL layer, which generates a strong signal at the air/ASL interface under x-ray phase contrast imaging (PCI) (Fig. 1b)[18, 19]. Because PCI does not allow us to observe the ASL/tissue interface, we used agarose beads instilled into the swine airway as 'measuring rods' to measure the ASL height by determining the position of the tissue with respect to the air/ASL interface (Fig. 1b, c, see "Methods" section). A researcher blinded to the experimental conditions measured ASL height as the distance between the air/ASL interface and the edge of the agarose bead touching the surface epithelium (Fig. 1b). The surface tension of the ASL immobilizes the beads onto the surface of the epithelium (see "Methods"). The liquid secreted by the airway, which would

normally be cleared out by mucociliary clearance, is retained around the static bead, allowing us to measure the accumulation of the ASL secreted by the airway[18]. Another advantage of agarose beads is that they can be used as a pathogen delivery system by loading the beads with CF relevant bacteria, such as *P. aeruginosa*.

**Absent ASL secretion in response to inhaled bacteria in CF**. We studied airway ASL secretory response to inhaled bacteria in anesthetized, 2–7-days-old CFTR$^{-/-}$ and wild-type swine. Our analyses indicate that the trachea of wild-type animals have a basal level of ASL secretion that increased by 100% after instillation of *P. aeruginosa*-laden agarose beads (Fig. 2a–c). In contrast, exposure to *P. aeruginosa*-laden beads failed to increase ASL secretion in CFTR$^{-/-}$ swine (Fig. 2d–f). These results show that ASL secretion is indeed upregulated by bacterial inhalation, revealing a previously unknown component of the airway's innate immune response. This response is absent in CF airways, which would result in reduced bacterial clearance, thus facilitating infection.

To understand the lack of response in CF lungs, we investigated the mechanism(s) underlying *P. aeruginosa*-mediated ASL secretory response. Pathogen detection in the airway requires the activation of pattern recognition receptors (PRRs), which in turn trigger more specific innate immune responses[20]. For instance, in CF15 cells, the response to *P. aeruginosa* is mediated by the activation of Toll-like receptor (TLR) 5 by the bacterial protein flagellin, and bacteria strains that lack flagellin (i.e., PAKΔfliC) fail to elicit any response[21]. Similarly, our results show that PAKΔfliC bacteria fail to trigger ASL secretion in vivo, whereas beads loaded with flagellin alone produced the same stimulatory effect as *P. aeruginosa* (Fig. 3a). These data suggest that airway response to *P. aeruginosa* is primarily mediated by stimulation of PRRs by flagellin. Because other relevant CF pathogens such as *Staphylococcus aureus* and *Haemophilus influenzae* do not express flagellin, we tested whether these pathogens stimulate ASL production. We found that *S. aureus*- and *H. influenzae*-laden beads also stimulated ASL secretion in wild-type swine (Fig. 3b). In addition, beads loaded with lipopolysaccharide (LPS), a common PRR stimulating endotoxin present in *S. aureus* and *H. influenzae*, also triggered similar ASL secretory responses in wild-type trachea (Fig. 3b). Interestingly, introducing beads loaded with the non-pathogenic bacteria *Escherichia coli* (DH5-α) and *Bacillus subtilis* failed to stimulate airway ASL secretory response (Fig. 3c). These data indicate that bacteria-triggered ASL secretion is an innate immune response specific to pathogens and occurs via the activation of TLRs, which would lead to the production of proinflammatory cytokines, e.g., interleukin (IL) 1β, TNFα, IL6, and IL8 that stimulate ASL secretion[15, 18, 22–24]. In CF airways, pathogen inhalation would stimulate TLRs and trigger proinflammatory signals[25]; however, IL1β, TNFα[15], IL6, and IL8 (Fig. 3d, f) fail to trigger ASL secretion and clear pathogens, thus facilitating infection and inflammation.

**Reduced basal ASL secretion in CF**. The ability of CF airways to clear pathogens may also be compromised by a reduction in basal ASL secretion, i.e., background ASL production independent of bacteria inhalation. Indeed, using bacteria-free agarose beads, we detected a basal ASL secretion rate both in wild-type and CFTR$^{-/-}$ swine in vivo, however, the basal ASL secretion by CFTR$^{-/-}$ swine was significantly smaller (Fig. 4a). Therefore, we decided to study the source of this secretion and its regulation. Our results show that basal ASL secretion is produced by submucosal glands, and surgical removal of over 90% of the

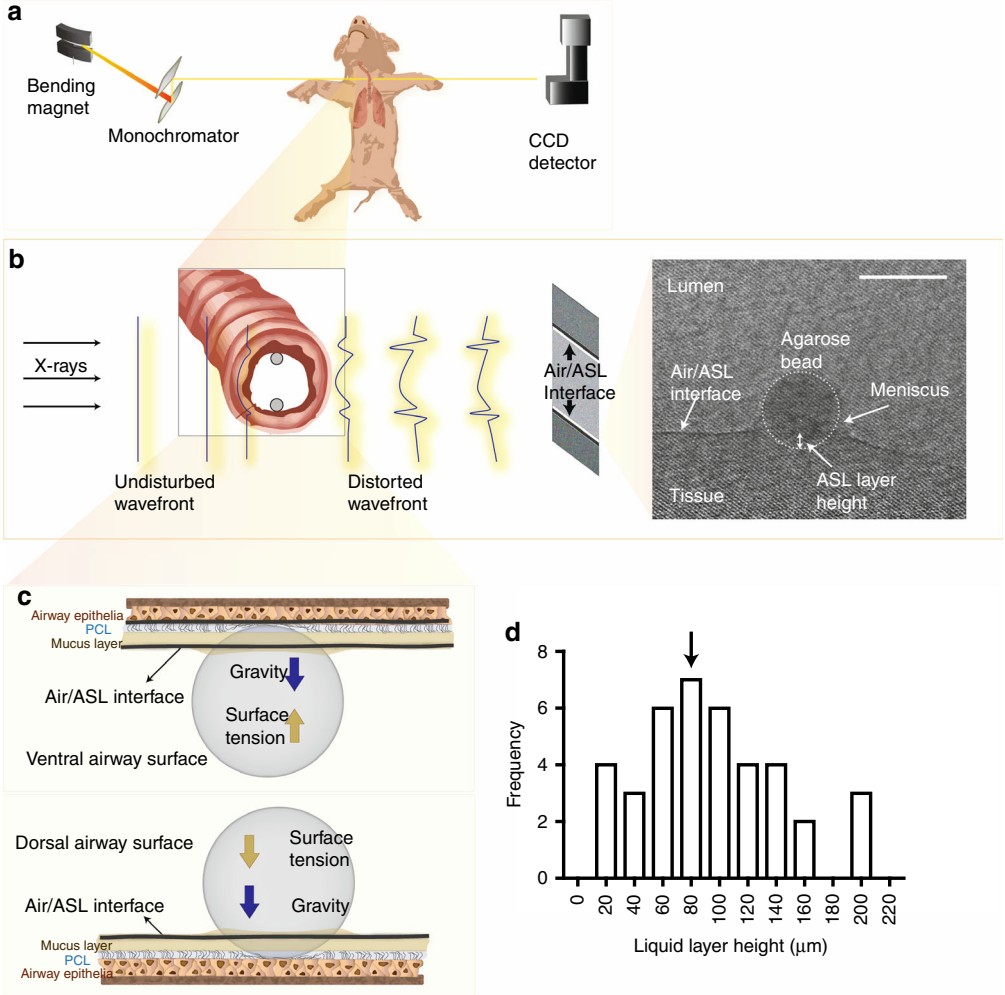

**Fig. 1** Synchrotron-based phase contrast imaging. **a** Schematic of the in vivo imaging set up. **b** Detection of the air/ASL layer interface in the lumen of swine trachea in vivo. When x-ray (undisturbed wavefront) pass through the trachea, the difference in refractive index between the ASL and the air in the lumen results in a phase shift of the x-ray (distorted wavefront), which is detected by phase contrast imaging (PCI). PCI cannot resolve the ASL/tissue interface because the ASL refractive index is very similar to that of the tissue. Thus we used agarose beads as "measuring rods" to determine the position of the tissue with respect to the air/ASL interface. In the phase contrast image obtained in vivo (grayscale image on *right panel*), the air/ASL interface is clearly observable as a *dark line* (highlighted with *dotted line*). The edge of the agarose bead is outlined by the *dotted circle*. The distance from the bottom of the agarose bead to air/ASL interface is used to measure the height of the ASL layer. Note that the air/ASL interface is below the level of the meniscus formed by the ASL on the agarose bead. *Scale bar* represents 500 μm. **c** Schematic diagram illustrating the forces acting on the agarose bead (surface tension and gravity) and the bead–tissue interaction. **d** Frequency distribution and mean (93 ± 8 μm, indicated by arrow) of ASL height in wild-type swine trachea exposed to bacteria-free agarose beads in vivo ($n = 39$ agarose beads obtained from 24 wild-type live pigs)

submucosal glands from wild-type ex vivo tracheal preparations significantly reduced ASL secretion (Fig. 4b). These results are consistent with previous reports that more than 90% of ASL produced in the upper airway is secreted by the submucosal glands[14]. Treatment of wild-type ex vivo tracheal preparations with the CFTR inhibitor, CFTRinh172, blocked basal ASL secretion (Fig. 4c), reinforcing that CFTR-mediated glandular secretion is the source of basal ASL production in the trachea.

It is possible that spontaneous ASL secretion could be a reactive response of sensory neurons to the presence of agarose beads. We tested this by blocking sensory neurons in ex vivo trachea preparations with the topical anesthetic lidocaine, which caused a slight reduction in basal secretion, but not to the extent that would explain loss of secretion in CF tissues (Fig. 4d). ASL secretion may also be stimulated by airway intrinsic neurons[13]. Thus, we treated preparations with tetrodotoxin, a $Na^+$ channel blocker that inhibits airway intrinsic neuron function[26], which

had no discernible effect on basal ASL secretion (Fig. 4e). We also tested whether parasympathetic tonic stimulation may drive basal secretion[27] by treating airways with the cholinergic antagonist, atropine. The treatment had no effect on secretion in vivo or ex vivo (Fig. 4f, g), indicating that basal secretion is independent of neuronal activation (i.e., parasympathetic, sensory efferent, and airway intrinsic neurons). However, blocking cAMP production with the adenylate cyclase blocker, SQ22536, significantly reduced basal secretion ex vivo (Fig. 4h), suggesting that a cAMP-mediated regulatory mechanism modulates basal secretion that is independent of the nervous system.

## Discussion
This study provides evidence of previously unknown components of the airway innate immune response to pathogens. We describe a basal level of ASL secretion that perhaps performs a

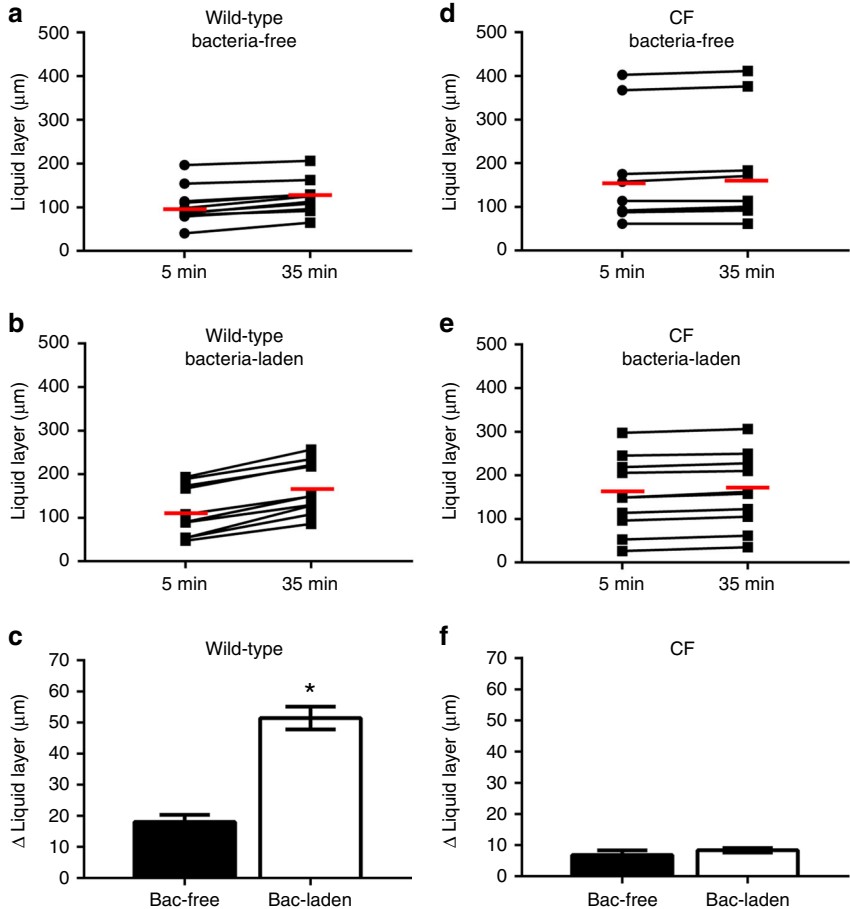

**Fig. 2** CFTR$^{-/-}$ swine fail to respond to *Pseudomonas aeruginosa* in vivo. Scatter plot and median (*red line*) of the effect of **a** bacteria-free and **b** bacteria-laden agarose beads on the ASL layer of wild-type swine. Bacteria-free beads trigger an ASL increase from 105 to 123 μm ($P < 0.05$, $n = 10$ from seven pigs, Wilcoxon match-pairs signed rank test) while bacteria-laden beads triggered a height increase from 117 to 168 μm ($P < 0.05$, $n = 10$ from six pigs, Wilcoxon match-pairs signed rank test). **c** Increase in ASL height triggered by bacteria-laden ($n = 10$) and bacteria-free ($n = 10$) agarose beads ($P < 0.0001$, $t = 7.76$, df = 18, Student's *t*-test). Effect of **d** bacteria-free and **e** bacteria-laden agarose beads on the ASL layer of CFTR$^{-/-}$ swine. Bacteria-free beads trigger an ASL increase from 172 to 178 μm ($P < 0.05$, $n = 9$ from six pigs, Wilcoxon match-pairs signed rank test) while bacteria-laden beads triggered a height increase from 150 to 158 μm ($P < 0.05$, $n = 11$ from eight pigs, Wilcoxon match-pairs signed rank test). **f** The increase in ASL height triggered by bacteria-laden ($n = 11$) and bacteria-free ($n = 9$) agarose beads were not significantly different in CFTR$^{-/-}$ swine ($P = 0.49$, Mann–Whitney test). Data are presented as mean ± SEM and within each panel the columns labeled with an *asterisk* differ significantly

"housekeeping" function[12] by ensuring a basal presence of mucins and antimicrobial compounds in the airway, which may serve a defensive role against inhaled pathogens and non-pathogens. In addition, the airway has a pathogen-triggered ASL secretion response at the site of infection through paracrine stimulation of ASL secretion by cytokines[6–8, 12, 15–17]. The lack of basal secretion in CF airway would result in reduced ASL availability to deal with inhaled particles, which combined with the lack of response to pathogens would severely undermine the innate immune system's ability to protect the airway from inhaled insults. Instead, pathogens would activate PRRs and production of proinflammatory signals, thus favoring inflammation[4, 5, 28]. Finally, our results suggest that treatments which improve the basal ASL secretion, such as hypertonic saline nebulization, or treatments that protect the airway from inhaled bacteria, such as prophylactic antibiotics, could slow the progression of CF airway disease.

## Methods

**Animals**. We used gut-corrected CFTR$^{-/-}$ swine purchased from Exemplar Genetics (Iowa, USA) and wild-type swine from Prairie Swine Center (University of Saskatchewan) that served as controls.

A sow implanted with cloned embryos with the gut-corrected CFTR$^{-/-}$ genotype (CFTR$^{-/-}$;TgFABP > pCFTR pigs)[29] was purchased from Exemplar Genetics. The sow was housed at the Western College of Veterinary Medicine Animal Care Unit, University of Saskatchewan, for 4 weeks before due date to allow the sow to acclimate to the new environment and reduce stress during delivery. To reduce exposure to bacteria in the birth canal and to minimize the stress of natural birth on the piglets, the delivery was done through a cesarean section performed by a specialized veterinary surgeon team to ensure minimal exposure of the piglets to bacteria.

The sow delivered 11 male piglets of normal weight (Supplementary Table 1). Each newborn animal was identified with a number and monitored 24 h a day. Food intake, urination, passing and consistency of feces, temperature, weight, respiratory function, O$_2$ saturation, body condition, and responsiveness were recorded for each piglet (Supplementary Table 1). Animal number 11 showed reduced O$_2$ saturation at birth that did not improve. Thus we decided to image it (this data was not included in our data set), and euthanized the animal 12 h after birth. Animals 3, 4, 5, 7, and 10 started to display deterioration of health status 48 h after birth, so we decided to image the animals and euthanized them. Animals 2, 6, and 8 displayed health deterioration 5 days after birth; thus, we imaged and euthanized the animals. Animals 1 and 9 reached the end of the experimental period at day 7 after birth. The control experiments were performed using 7 days old wild-type swine 3– 5 kg in weight (Prairie Swine Center, University of Saskatchewan). For ex vivo experiments, tracheas from CFTR$^{-/-}$ and wild-type animals where dissected within 15–30 min after euthanasia. The trachea were clamped and placed in ice-cold Krebs-Ringer solution (mM): 115 NaCl, 2.4 K$_2$HPO$_4$, 0.4 KH$_2$PO$_4$, 1.2 CaCl$_2$, 1.2 MgCl$_2$, 25 NaHCO$_3$, 10 glucose (pH = 7.4) equilibrated with 95% O$_2$, 5% CO$_2$ until used. All experiments were performed

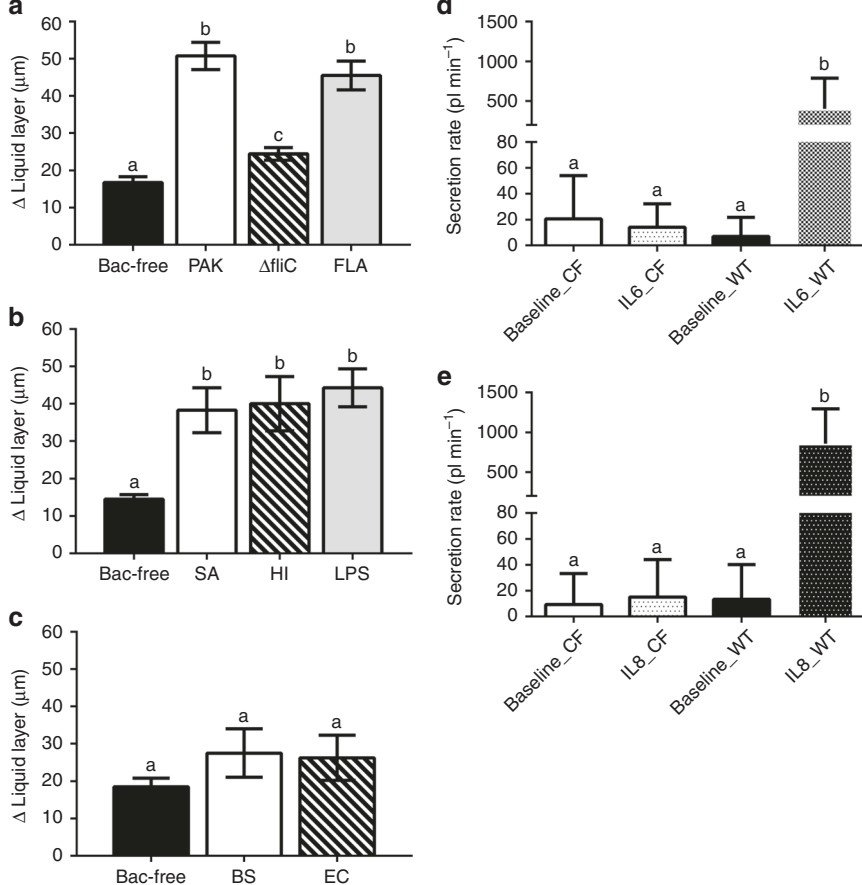

**Fig. 3** Stimulation of pattern recognition receptors triggers ASL secretion in wild-type swine in vivo. **a** *P. aeruginosa* PAK strain-laden agarose beads (PAK) significantly stimulates ASL secretion. PAKΔfliC-laden agarose beads (ΔfliC), which lacks flagellin, induce much less ASL secretion. Flagellin-laden (10 ng/ml) agarose beads (FLA) triggered a similar response as PAK beads ($n = 11$ bacteria-free from six pigs, $n = 10$ PAK from six pigs, $n = 10$ ΔfliC from seven pigs, and $n = 8$ FLA from five pigs; $P < 0.0001$, $F(3, 35) = 36.59$, ANOVA and Tukey's multiple comparison test). **b** *S. aureus*-(SA), *H. influenza*-laden (HI), and LPS (20 μg/ml)-laden agarose beads stimulate ASL secretion in wild-type swine ($n = 10$ bacteria-free from seven pigs, $n = 7$ *S. aureus* from four pigs, $n = 7$ *H. influenzae* from four pigs, and $n = 7$ LPS from five pigs; $P = 0.0003$, $F(3, 27) = 8.79$, ANOVA and Tukey's multiple comparison test). **c** The non-pathogenic *B. subtilis* (BS)- and *E. coli* (EC)-laden agarose beads failed to trigger ASL secretion in wild-type swine ($n = 10$ bacteria-free from six pigs, $n = 7$ *E. coli* DH5-α from five pigs, and $n = 7$ *B. subtilis* from six pigs; $P = 0.34$, $F(2, 21) = 0.40$, ANOVA and Tukey's multiple comparison test). Secretion assay data showed that **d** IL-6 (50 ng/ml, $n = 16$ preparations from seven CFTR$^{-/-}$ animals and $n = 9$ preparations from four wild-type animals, $P < 0.0001$, $F(3, 60) = 12.79$, ANOVA and Tukey's multiple comparison test) and **e** IL-8 (50 ng/ml, $n = 15$ for CFTR$^{-/-}$ airway tissue from seven pigs and $n = 10$ for wild-type tissue from four pigs, $P < 0.0001$, $F(3, 56) = 13.09$, ANOVA and Tukey's multiple comparison test) both trigger submucosal gland secretion in wild-type airway, but fail to stimulated ASL secretion in CF airway. Data is presented as mean ± SEM and within each panel the columns labeled with *different letters* differ significantly

within 18 h after euthanasia. All of the experiments were performed under the approvals of the Canadian Light Source and the Animal Ethics Committee of the University of Saskatchewan.

**Analysis of CFTR expression in the gut**. The CFTR expression correction in the gut of gut-corrected CFTR$^{-/-}$ piglets was tested by quantitative polymerase chain reaction (qPCR) analysis[29]. Gut tissue (from pig #7 and a wild-type pig) was dissected, flash frozen in liquid nitrogen, and stored at −80 °C. The samples were later thawed in RNA-Ice (Ambion, cat. no. AM7030) overnight. Approximately 120–150 mg of the thawed tissue was used to extract total RNA using the Purelink RNA mini kit (Ambion, cat. no. 12183025). RNA quality was checked on a BioAnalyzer (Agilent) to ensure that RNA templates used in downstream applications had a RNA integrity number (RIN) > 8.0 and was not contaminated with genomic DNA. First strand cDNA was synthesized from ~1 μg of total RNA using random primers provided in the Superscript IV First Strand Synthesis Kit (ThermoFisher, cat. no. 18091050). Quantitative reverse transcription polymerase chain reaction (RT-qPCR) reactions were carried out using Taqman expression assays and probes (CFTR: Ss3389420_m1, beta actin: Ss03376160_u1, Thermo-Fisher) previously used to measure CFTR mRNA levels in the CFTR$^{-/-}$ pigs[1]. qPCRs were done on a CFX96 thermocycler (BioRad) and analyzed using the CFX manager software. Samples were run in quadruplicates. qPCR results showed that CFTR mRNA levels (normalized to *β-actin*) in the gut of pig #7 was 23.29% that of

wild type (Supplementary Fig. 1), which is consistent with what has been previously reported[29].

**Pathological analysis of CFTR$^{-/-}$ swine**. After euthanasia, the carcasses were subjected to pathology analyses by a board-certified veterinary pathologist. Gross examinations were conducted on ten carcasses. Tissues included brains (pigs #4, 5, 7, 10, and 11 only), eyes (pigs #4, 5, 7, 10, and 11, only), hearts, spleens, kidneys, livers, and pancreas (pigs #8 and 11 only), small and large intestines were fixed in 10% formalin for haematoxylin and eosin staining and histological examination.

On gross examination (Supplementary Table 2), the colon and cecum in 7 out of 10 pigs were distended by feces that was firmer than normal (Supplementary Fig. 2a). The CFTR$^{-/-}$ animals with healthy guts (#1, 7, and 9) displayed the same abnormal response to inhaled bacteria phenotype as those with gut problems (data not shown). Four pigs that were euthanized later in the experiment showed cranioventral consolidation of the lung. Four pigs had gallbladders that were small (hypoplasia) on gross examination. In pig #10, the common bile duct was blocked by a thick mucus plug. Two pigs had increased amount of mucus in their nasal cavities compared to the other pigs (Supplementary Fig. 2b). On histological examination, the pancreases were hypoplastic (Supplementary Fig. 2c). Three pigs had evidence of cholestasis. The pneumonia was characterized by severe infiltration of neutrophils mixed with many bacteria in the alveoli and bronchiole

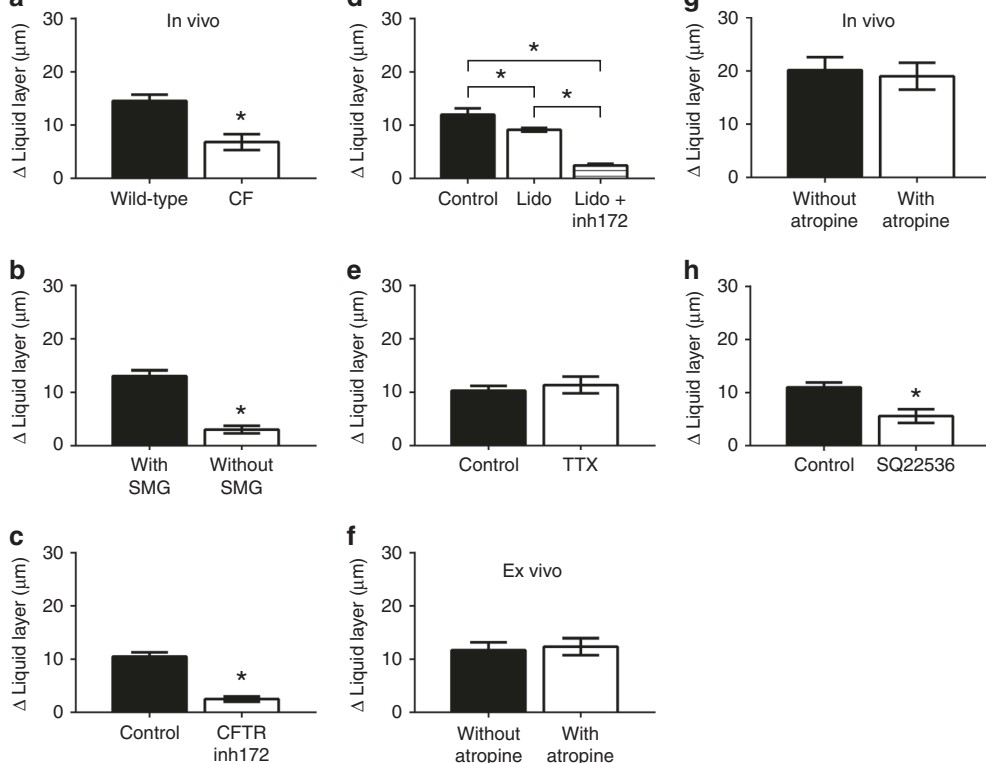

**Fig. 4** CF tissues fail to produce basal ASL secretion. **a** CFTR$^{-/-}$ swine (CF, $n = 9$ from six animals) display lower basal ASL secretion than wild-type in vivo ($n = 10$ from seven animals, mean ± SEM, $P = 0.001$, Mann–Whitney test). **b** Wild-type ex vivo preparations with surgically removed submucosal glands (SMG) produced lower basal ASL ($n = 12$ with SMG from six preparations and $n = 13$ without SMG from six preparations, $P < 0.0001$, Mann–Whitney test). **c** Wild-type ex vivo preparations incubated with CFTRinh172 (100 µM) produced lower basal ASL than non-treated preparations ($n = 6$ from four preparations for control, and $n = 14$ from five preparations for CFTRinh172, $P < 0.0001$, Mann–Whitney test). **d** Topical application of lidocaine (Lido) had a small effect on basal ASL secretion. Simultaneous treatment with the CFTR inhibitor (Lido + inh172) completely blocked basal ASL secretion ($n = 13$ control from five preparations, $n = 21$ Lido from eight preparations, and $n = 16$ Lido + inh172 from seven preparations; $P < 0.0001$, $F(2, 47) = 59.35$, ANOVA and Tukey's multiple comparison test). **e** Tetrodotoxin (1 µM, TTX) treatment had no effect on basal ASL secretion ($n = 18$ control from eight preparations, and $n = 11$ TTX from four preparations, $P = 0.58$, Mann–Whitney test). Atropine had no effect on basal ASL secretion in **f** ex vivo preparations (10 µM, $n = 8$ without atropine from three preparations and $n = 8$ with atropine from four preparations, $P = 0.76$, $t = 0.30$, df = 14, Student's $t$ test) or **g** in vivo wild-type swine (treated with 0.04 mg/kg IM 2–10 min following induction of anesthesia, $n = 8$ without Atropine from six animals, and $n = 10$ with Atropine from seven animals, $P = 0.75$, $t = 0.32$, df = 16, Student's $t$ test). **h** Treatment with SQ22536 (0.5 mM) blocked ASL secretion ($n = 10$ control from four preparations and $n = 11$ SQ22536 from four preparations; $P = 0.0003$, Mann–Whitney test). Data is presented as mean ± SEM and within each panel the columns labeled with an *asterisk* differ significantly

(Supplementary Fig. 2d). Brains, eyes, hearts, spleens, kidneys, small and large intestines were normal histologically.

There was no evidence of septicemia (septic shock) on gross or microscopic examination in any of the animals studied. Septicemia in pigs would usually display petechial or ecchymotic hemorrhage in subcutaneous tissues, serosal surfaces of different organs, and renal parenchyma. We would also expect to observe pulmonary edema, and gastric fundic congestion. None of these gross anatomical lesions were observed in any of the animals. Septicemia would also cause microscopic lesions including hemorrhage and fibrin thrombi in different organs, especially in the stomach and lungs, as well as cause bacterial emboli in various organs. None of the above changes were observed, thus, we concluded that septicemic shock was unlikely in any of the animals used in this study.

**X-ray imaging**. Experiments were performed at the BioMedical Imaging and Therapy (BMIT) facility, Canadian Light Source (CLS), Saskatchewan, Canada[30]. All imaging was done using the Bending Magnet (BM) beamline 05B1-1 endstation. The experimental hutch was located 25.5 m from the storage ring. Monochromatic x-rays (33.5 keV, $\lambda = 0.037$ nm for live swine experiments, and 20 keV, $\lambda = 0.062$ nm for isolated trachea experiments) were selected using a standard double-crystal monochromator. At the imaging station, the beam size was ~100.0 mm horizontal × 6.0 mm vertical. A propagation (sample-to-detector) distance of ~115 cm (in vivo) and ~65 cm (ex vivo) were chosen to enhance the air/ASL layer interface. Images were captured using a charge-coupled device (CCD) detector (C9300-124 Hamamatsu Photonics) with a high resolution X-ray converter (AA-60 Hamamatsu Photonics). The pixel size of the captured image was

8.75 × 8.75 µm. The exposure time for each image ranged from 400 to 600 ms depending on the storage ring current so as to maximize signal-to-noise ratio for image capturing.

**Evaluation of agarose beads–epithelium interaction**. We used agarose beads as "measuring rods" to determine the height of the ASL layer. ASL height was measured as the distance between the air/ASL interface and the edge of the agarose bead touching the surface epithelia. The validity of our measurement relies on the fact that the agarose beads instilled in the ASL do indeed come into physical contact with the airway surface epithelium and are immobilized[18]. We therefore conducted a series of experimental tests and theoretical analysis to determine whether the beads are in fact immobilized on the surface of the airway epithelia.

Agarose beads immersed in the ASL are subjected to two major forces that determine the stability of the beads: gravity and the surface tension created by the ASL. At the dorsal (bottom, in supine position) surface of the trachea, both gravity and surface tension force the beads towards the surface epithelium (Fig. 1c); as a result, the beads are forced against the airway surface epithelium as long as the bead diameter is larger than the ASL thickness. For the beads placed on the ventral (top) surface of the trachea, gravity acts to pull the bead away while ASL surface tension forces the bead towards the surface epithelium. In this case, the net force on the bead determines whether the bead would come into contact with the airway surface epithelium. We thus produced a mathematical model of the bead–ASL/epithelium interaction to determine whether the beads were in contact with the epithelium under our experimental conditions.

 

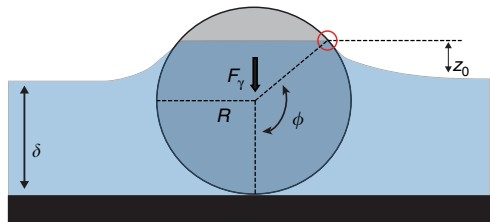

**Fig. 5** Diagram of an agarose bead immersed in ASL. The diagram shows the force due to surface tension and the capillary forces acting on the bead, and the parameters relevant to Eq. (3). Where $R$ is the radius of the agarose bead, $\gamma$ is the surface tension of ASL, $\delta$ is the height of the ASL layer in the trachea, $z_0$ is the meniscus depth of the ASL due to the capillary effect of liquid around the agarose bead, $\theta$ is the contact angle between the ASL and the agarose. Our analysis shows that under our experimental conditions, the net effect of gravity and surface tension push the beads against the epithelium without causing significant tissue displacement

The adhesion force due to surface tension and capillary forces on the bead (Fig. 5) can be calculated as[31, 32]:

$$F_\gamma = 2\pi R\gamma \sin\phi \sin(\theta + \phi). \tag{1}$$

Where $R$ is the radius of the agarose bead, $\gamma$ is surface tension of ASL, $\delta$ is the height of the ASL layer in the trachea, $z_0$ is the meniscus depth of ASL due to the capillary effect of liquid around the agarose bead, and $\theta$ is the contact angle between ASL and agarose.

$$\cos\phi = 1 - \frac{\delta}{R} - \frac{z_0}{R}. \tag{2}$$

From Eqs. (1) and (2):

$$F_\gamma = 2\pi R\gamma \sin\left[\cos^{-1}\left(1 - \frac{\delta + z_0}{R}\right)\right] \sin\left[\theta + \cos^{-1}\left(1 - \frac{\delta + z_0}{R}\right)\right]. \tag{3}$$

To resolve Eq. (3), we obtained airway fluid surface tension values ($\gamma$), previously reported to range from 30 to 34 mN/m[33]. We used the average value of 32 mN/m in our calculations. We measured the contact angle ($\theta$) between the agarose and the ASL using a goniometer[34]. A droplet of swine trachea ASL was placed on a flat surface of 4% agarose, and the contact angle between the ASL and the 4% agarose was directly recorded to be 44° ± 1.6°.

Inputting these values to Eq. (3), we obtain:

$$F_\gamma = 2\pi R \cdot 32\text{mN/m} \cdot \sin\left[\cos^{-1}\left(1 - \frac{\delta + z_0}{R}\right)\right] \sin\left[44° + \cos^{-1}\left(1 - \frac{\delta + z_0}{R}\right)\right]. \tag{4}$$

We can assume for now that $z_0$ will be small compared to the thickness (averaging 93 μm) of the fluid layer. However, the thinner the ASL is, the more important $z_0$ becomes. And larger $z_0$ value would cause larger surface tension on the bead. Since $z_0$ would have minimal effect in our case, we simplified the force calculation to Eq. (5), where we underestimate the surface tension generated by ASL.

$$F_\gamma = 2\pi R \cdot 0.032\text{N/m} \cdot \sin\left[\cos^{-1}\left(1 - \frac{\delta}{R}\right)\right] \sin\left[44° + \cos^{-1}\left(1 - \frac{\delta}{R}\right)\right]. \tag{5}$$

Now there are only two variables; bead radius, $R$, and ASL layer height, $\delta$.

Supplementary Fig. 3 shows the change in adhesion force, within the range of ASL thickness observed in swine trachea, 20–300 μm, with varying bead size and fluid thickness. The data show that negative adhesion, i.e., the bead would move away from tissue, occurs when the bead is fully submerged in the fluid. In that condition, the surface tension force would be considered nil, and gravity, $G = 4/3\rho\pi R^3$, would force the bead away from the epithelium. The results show that beads with a range of diameters used in our experiments, 400–1000 μm, will touch the epithelium even with an ASL height as little as 2 μm (Supplementary Fig. 3). Thus, under all our experimental conditions, the net effect of gravity and surface tension push the beads against the epithelium.

We also tested whether the forces acting on the agarose beads were strong enough to push the beads into the airway surface epithelium and deform the tissue, which would affect our measurements. The deformation of the trachea surface epithelia caused by the force applied on the tissue is related to the elasticity of the tissue[35].

We measured the elasticity of swine trachea by measuring the amount of force required to displace the tracheal tissue[35, 36]. The forces that cause varying displacement for trachea fall into a linear function, and the measured spring constant κ of swine trachea was 7.6 N/m. The displacement of tracheal epithelium by beads of varying sizes and and varying ASL layer thickness can be calculated

from Eq. (6).

$$d = F_{\text{net}}/\kappa. \tag{6}$$

Where $d$ is the displacement of trachea tissue, $F_{\text{net}}$ is the net force on the agarose bead (for the beads at top of trachea, $F_{\text{net}} = F_\gamma - G$; for the beads at bottom of trachea, $F_{\text{net}} = F_\gamma + G$), and κ is the spring constant of swine trachea we measured. The results indicate that displacement of airway epithelia by agarose beads would not be larger than 12 μm (Supplementary Figs. 4 and 5), which is within the range of the error of our measurments. Therefore, tissue displacement is unlikely to affect our measurements of ASL height.

We conducted experimental tests to further study whether the agarose beads in our experiments are in contact with the surface epithelia as expected based on our theoretical model. Our results show that, as indicated by our mathematical model, the beads indeed contact the epithelia. Initially, we placed agarose beads in an ASL sample collected from isolated trachea and placed it on the underside of a glass slide. The agarose beads were forced against the glass surface, indicating that the surface tension from the ASL is stronger than gravity and forces the beads up against the glass surface, as predicted by the mathematical model (Supplementary Fig. 6).

Histological evidence also supports the fact that the agarose beads are in direct contact with the surface epithelia. Agarose beads were instilled into freshly harvested trachea of 1 week old wild-type piglets. The preparations were flash frozen in liquid nitrogen and stored at −80 °C. Then, the trachea preparations were cryo-sectioned into 20 μm sections and stained with Nuclear Fast Red (Sigma) for 1 min, which delineated the airway tissue. The sections were imaged on a light microscope. Histological images ($n = 11$) clearly showed that the agarose beads were indeed in contact with the airway surface epithelium (Supplementary Fig. 7), consistent with published two-photon microscopy images showing that the beads are in direct contact with the epithelia[18].

We also tested the prediction that tissue displacement by beads does not affect our measurements of the ASL height. First, if displacement of the tissue by beads did affect our ASL height measurements, one would expect that the ASL heights measured from beads located at the top part of the trachea, where gravity and surface tenssion act in opposite directions, would be smaller than those obtained from beads located at the bottom part of the trachea, where both gravity and surface tesnion push the beads into the tissue. Our analysis of the data collected shows that ASL measurements obtained from top and bottom of the trachea were not significantly different (Supplementary Fig. 8).

In addition, since the tissue displacement depends on bead size (Supplementary Figs. 4 and 5), we would expect that, if tissue displacement affects our measuremetns, bead size would influence the ASL hight measured. A small bead would cause less tissue displacement than a larger bead. As such, we should expect a correlation between bead size and the ASL height measured (Supplementary Fig. 9) or the change in ASL in 30 min (Supplementary Fig. 10). However, our data showed no such correlations between ASL height or change in ASL and bead size. Indicating that tissue displacement does not affect our ASL measurements.

Thus, using a range of theoretical and experimental evidence, we confirm that beads instilled in the ASL experience sufficient surface tension force that immobilize them against the airway, without significantly deforming the epithelium or affecting ASL measurement.

**Preparation of agarose beads**. Agarose beads were made in sterile conditions with 4% agarose in PBS[37, 38]. Insoluble $BaSO_4$ or CuI (1 M) was loaded into the agarose beads to make them visible in x-ray imaging without affecting the osmotic pressure of the beads[18]. Bacteria-free agarose beads were made from sterile PBS alone; and bacteria-laden agarose beads were made by adding 10% v/v of bacterial stock solution.

*P. aeruginosa* stock solution was made with a clinical isolate, *P. aeruginosa* (NH57388)[39], generously provided by Dr. John Gordon at the University of Saskatchewan. Clinical isolates of *H. influenzae* and *S. aureus* were generously provided by Dr. Joseph Blondeau at the Royal University Hospital, Saskatoon, Saskatchewan, Canada. *E. coli* DH5-α was provided by Dr. Gordon at the University of Saskatchewan, and *B. subtilis* was purchased from Cedarlane (Burlington, Ontario, Canada).

Aliquots of the *P. aeruginosa* clinical isolate, the *P. aeruginosa* PAK and PAKΔfliC strains, *S. aureus* clinical isolate, and *E. coli* DH5-α were inoculated in Erlenmeyer flasks containing growth media, and incubated for ~24 h at 37 °C with shaking at 170 rpm in an aerobic environment. An aliquot of the *H. influenzae* clinical isolate was incubated for ~24 h at 37 °C with shaking at 170 rpm in an anaerobic environment, and an aliquot of *B. subtilis* was incubated for ~24 h at 26 °C with shaking at 170 rpm in an aerobic environment. All bacterial cultures except *E. coli* and *B. subtilis* were then heat killed (by incubating at 80 °C for 30 min), concentrated by centrifuging, and then stored at − 20 °C. Prior to use, pathogen concentrates were thawed and mixed with agarose; whereas non-pathogens (*E. coli* DH5-α and *B. subtilis*) were directly concentrated from bacterial culture aliquots before use.

**Experimental set-up**. For the in vivo experiments, the animal was placed in supine position on a warming pad and anesthetized with 2–2.5% isoflurane in pure

medical $O_2$ through a face mask at an air flow rate of 1 l/min. Throughout the experiment, the animals breathe spontaneously. Respiratory rate, heart rate, body temperature, and $O_2$ saturation level as well as the plane of anesthesia were monitored. The agarose beads with diameters ranging from 400 to 1000 µm where instilled into the trachea of the animal using an endotracheal tube. Before inserting the endotracheal tube, the larynx was sprayed with lidocaine to prevent a reflex response to intubation. The agarose beads were then blot-dried and placed in the endotracheal tube. The endotracheal tube was placed at the opening of the larynx into the trachea and the beads were blown with a puff of air out of the tube and into the trachea, after which, the endotracheal tube was immediately removed. Images were taken at 5 and 35 min after the agarose beads were placed in the trachea.

For ex vivo experiments[18], tracheas were dissected from recently euthanized animals and placed in a custom-built chamber wherein the tissue was immersed in Krebs solution plus 1 µM indomethacin at 35 °C and equilibrated with 95% $O_2$ and 5% $CO_2$. The trachea was sealed so that the lumen remained free of the solution but it was accessible for introduction of beads. Agarose beads ranging ~400–1000 µm in diameter were blotted dry and placed in the lumen of the airway.

The air-ASL interface can be detected only at the ventral (top) and at the dorsal (bottom) of the trachea, where the direction of the x-rays propagation is parallel to the plane of the air-ASL interface because only these regions generate x-ray phase shifts that produce a strong signal in phase contrast x-ray imaging (Fig. 1b). We therefore used a computer-controlled motorized stage to rotate the sample to ensure the beads were positioned appropriately before imaging. Since we had no prior knowledge of the exact position of the agarose beads, we rotated the sample (i.e., the anesthetized swine or isolated trachea) along its longitudinal axis 1° at a time and recorded an image. We repeated this process for 23°. The ASL height was measured using those beads that reached the top or the bottom of the trachea, at which point the Air/ASL interface becomes visible.

The ASL height was measured 5 and 35 min after instillation of the beads. The measurements made at 5 min were interpreted as the original state of ASL, e.g., before exposure to bacteria. We labeled the first time point as 5 min (instead of 0) to reflect the time delay between instillation of the beads and acquisition of the first image.

Both bacteria-free and bacteria-laden beads were instilled in each experiment (in vivo and ex vivo) to control for differences in response among preparations. We distinguished bacteria-laden beads from bacteria-free beads using K-edge subtraction that allows us to determine iodine from barium labeled beads[40]. Iodine K-edge subtraction imaging was performed using monochromatic 33.20 and 33.35 keV beams. Barium K-edge subtraction imaging was performed using monochromatic 37.50 and 37.65 keV beams.

**Secretion assays**. A sample of trachea was cut along the trachealis muscle, and the airway submucosa containing the glands was dissected from the cartilage[41]. The submucosa preparation was then placed in a custom-built chamber with the serosal side of the preparation bathed in Krebs solution containing 1 µM indomethacin. The preparation was maintained at 37 °C and equilibrated with warm, humidified 95% $O_2$ and 5% $CO_2$ gas (TC-102, Medical Systems Corp., Greenvale, NY, USA). The mucosal side was cleaned and dried with a stream of air, and then coated with ~5 µl mineral oil. The ASL secreted by the submucosal glands formed a spherical droplet under the mineral oil. Cytokines IL6 or IL8 were added to the serosal side. The droplets of ASL secreted by the glands were imaged every 120 s. Image data files were stored for offline analysis using ImageJ 1.43u (NIH). The volumes of the secreted droplets were calculated assuming a spherical shape ($V = 4/3\pi r^3$)[41]. The secretion rate was calculated by fitting the volume versus time plots with straight lines using linear regression, and the slopes were taken as the secretion rates using a minimum of four points[42].

**Surgical removal of the submucosal glands**. The cartilage was removed from a tracheal preparation using a scalpel and a blunt-ended elevator[18]. Once the cartilage was removed, the submucosal glands and other tissue in the submucosa were dissected out by "shaving" the surface using a breakable scalpel blade (Fine Science Tools, Vancouver, Canada). Using chamber and secretion assay experiments showed that our procedure allowed us to remove 90% of glands from the trachea without damaging the surface epithelia[18].

**Reagents**. CFTRinh-172, SQ22536, and flagellin from *Salmonella typhimurium* were obtained from Cedarlane Labs (Burlington, ON, CA). Atropine and lipopo-lysaccharide from *P. aeruginosa* were purchased from Sigma-Aldrich; tetrodoxin was purchased from Alomone labs (Jerusalem, Israel); lidocaine hydrochloride 12 mg per metered dose was obtained from Odan Laboratories Ltd. (Montreal, Canada). Stock solutions of CFTRinh-172, SQ22536, and were dissolved in DMSO. The final concentration of DMSO was < 0.1%. Atropine, flagellin, and lipopoly-saccharides were dissolved in milli-Q water.

**Statistics**. Data are presented as mean ± SEM. The data sets that met the assumption of normality and homogeneity of variances where analyzed with parametric tests, ANOVA or Student's t-test. For those that did not, we performed non-parametric tests, Wilcoxon match-pairs signed rank test or Mann–Whitney

U-test, using GraphPad Prism 5 (GraphPad Software Inc., San Diego, CA, USA), and $P < 0.05$ was considered significant. Preliminary experiments showed that a sample size of 6 was sufficient to ensure adequate power to detect the effects of treatment on ASL secretion. Animal #11 was not included in the study because it never recovered from the stress of delivery. We did not exclude any data collected from any of the animals studied from our analysis.

**Data availability**. The authors declare that all the data supporting the findings of this study are available within the article and its Supplementary Information files.

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

## Acknowledgements

We wish to thank Drs Robert Lamb, Ning Zhu, Adam Webb, and Tomasz Wysokinski for assistance in developing the synchrotron-based imaging technique and Drs Melanie Gibbons and Deborah Haines for their help with veterinary care of the animals. The BMIT facility is supported by the Canada Foundation for Innovation, Natural Sciences and Engineering Research Council of Canada, the University of Saskatchewan, the Government of Saskatchewan, Western Economic Diversification Canada, the National Research Council Canada, and the Canadian Institutes of Health Research. This work was funded by a Cystic Fibrosis Canada grant and a Canadian Institutes of Health Research grant to J.P.I. and a CIHR-THRUST scholarship to X.L.

## Author contributions

X.L.: developed experimental techniques, conducted the experiments, analyzed data, interpreted data, contributed to writing the manuscript. G.B.: developed experimental techniques, analyzed data. J.S.T.: interpreted data, contributed to writing the manuscript. S.J.: performed experiments, interpreted data, contributed to writing the manuscript, created artwork. N.H.: member of the animal husbandry team, conducted ASL measurements while blinded to the experimental conditions. P.G.: performed experiments, member of the animal husbandry team, conducted ASL measurements while blinded to the experimental conditions. N.G.: member of the animal husbandry team. Y.H.: contributed pathology analysis. J.L.C.: member of veterinary team, performed cesarean section. T.D.: member of veterinary team, performed anesthesia during cesarean section. T.J.: member of veterinary team, performed anesthesia during cesarean section. B.M.: developed husbandry protocols, member of the animal husbandry team. M.B.: developed husbandry protocols, member of the animal husbandry team. T.S.: produced mathematical model. O.Y.: conducted ASL measurements while blinded to the experimental conditions. V.A.C.: performed experiments, interpreted data, contributed to writing the manuscript. T.E.M.: contributed to research conception and design, developed experimental design, interpreting the data, contributing to writing the manuscript. L.D.C.: developed experimental techniques, analyzed data, contributed to writing the manuscript. J.P.I.: conceived of and designed the research study, designed and developed experimental techniques, interpreted the data, wrote the manuscript.

## Additional information

**Competing interests:** The authors declare no competing financial interests.

