## [Peer Review file · Nature Communications]

Reviewers' comments:

Reviewer #1 (Remarks to the Author):

The manuscript by Luan et al employ a novel in vivo imaging method (synchrotron-based x-ray imaging) to probe mechanisms of airway secretion in wild type and cystic fibrosis (CF) pigs. Using this model, they demonstrate that CFTR-dependent basal and toll-like receptor-mediated airway surface liquid (ASL) secretion occur in response to bacterial exposure (or bacterial components) and that this mechanism fails in CF pig trachea. These experiments attempt to fill in gaps in knowledge regarding how the CFTR-deficient airway differs from normal at homeostasis and when perturbed by daily bombardment of inhaled bacteria. This nicely complements and expands upon prior in vitro experiments conducted in human polarized ALI published by the UNC Chapel Hill group (Button et al, *Sci Signaling* 2013), and links that work to other aspects of the emerging model of impaired host defenses in the CF airway. However, the findings in this manuscript also contradict studies in the CF pig model by the Welsh group that demonstrate no abnormalities in ASL height in vivo (*Cell*. 2010 143:911-23). Nonetheless, other groups such as Rowe et al. show that ASL height in excised CF pig tracheal is lower than wild type using microoptical coherence tomography (*Am J Respir Crit Care Med*. 2014 190:421-432). Regardless of the CFASL debate, the strength of science in this manuscript pertains to the induced secretory response by bacterial pathogens and the new in vivo method for assaying ASL height in vivo.

There are few studies in CF pigs by groups outside University of Iowa, so I do think this adds to the significance of this manuscript and also points out limitations of the gut-corrected pig model that are not evident in previous publications. Given the debate on ASL height in CF pigs, this study also favors the prevailing model in the field of airway dehydration in CF, which is useful.

Major concerns:

1. There is one major scientific variable that could significantly impact the outcome of experiments. Table S1, and the paragraph in which it is cited, reports the health problems with the piglets. Initial reports of the model suggest that meconium ileus typically leads to septic shock by 48 hrs. This present study reports that gut-correction does not completely rescuing these gut related health problems. Seven of the eleven CF animals had intestinal complication and thus were likely septic at the time of their health failure. I am not sure what happens to the activation of toll-like receptors on the basal lateral membrane of airway epithelia, but it could certainly down-regulate the receptor from the apical surface of the epithelia. So the question is, does sepsis interfere with airway secretion and/or bacterial responses that promote airway secretions. The only way I can think to rule this out is to make WT pigs septic and repeat the studies. If secretions are reduced, then much of what they might be studying is a septic response in the lungs. This would still be interesting, but a different conclusion of their in vivo data.

2. Do they have evidence that dissecting away submucosal glands, which reduced basal secretions, does not damage the surface airway epithelium and its ability to secrete fluid?

Minor concerns:

1. Figure 3e is called 3f in the legend. In the text describing panels d and e, they don't explicitly state that IL6 and IL8 stimulate secretions in WT, they just state it does not do so in CF. It would be good to reword this to point out both aspects.

2. In generally the Figure legends are pretty wordy since they repeat all the statistical tests for each panel. It might be easier to read if they grouped the statistical test at the end of the legend for various levels of significance and if the N varies, they could write that next to the genotype below the x-axis.

3. It might be useful to point out that piglets were delivered by c-section to avoid bacterial exposure to the lung during birth. Presumably this is why they did that.
4. Since this is one of the only published studies in the gut-corrected pigs outside the Iowa group and this present study evaluated gut expression of the CFTR transgene, if they have the tissue it would be useful to include CFTR mRNA expression studies in the airways of CF pigs to confirm it is lower than WT and that the transgene is not ectopically expressed.
5. They excluded one CF piglet that was particularly affected with respiratory failure from birth (suppl material, line 14). However, they also state that they did not exclude any data (main body, line 288). The statements should be consistent.

Reviewer #2 (Remarks to the Author):

This study addresses important questions, whether the inhalation of bacteria triggers airway surface liquid (ASL) secretion and whether this process is abnormal in cystic fibrosis (CF). The authors developed a new technique to approach these issues. However, I have several concerns and question regarding the manuscript:

- The authors noted "The main challenge in testing this hypothesis is the lack of an imaging method.....", it is not clear why a highly specialized synchrotron-based x-ray imaging technique is necessary to measure the ASL layer, instead of a more common, commercially available, portable x-ray micro CT system applicable for live animals, which has sufficient spatial resolution. The authors need to clarify this.
- The authors stated that "the air-ASL interface can be detected only at the ventral (top) and at the dorsal (bottom) of the trachea.....", therefore, the animal (for in vivo imaging) or the harvested trachea (for ex vivo) was rotated to ensure that beads were positioned exactly at the top and at the bottom of the trachea using a computer-controlled motorized stage. The placement of the beads to precise positions (exactly ventral and dorsal) must be critical for the experiment and needs to be done in micron precision. Please provide more detailed description of this procedure.
- Is placing the plane of the x-ray exactly at the middle of a bead (Figure S3) crucial for applying the mathematical model the authors use? If so, please provide more detailed description of how this is done (the experimental procedure).
- Since it is crucial for the measurement that the agarose beads are immobilized on the surface of the epithelium due to the surface tension of the ASL, a video demonstration as a supplementary material would be nice. Although the authors state this on lines 86-90 in the text, it would be easier for the reader to see it as a video.
- The authors use the average value of 32 mN/m of airway fluid surface tension (γ) in their calculation, is this value reasonable in the case of CF?
- Does the viscosity of ASL change in patients with CF? If it does, can the surface tension of ASL be considered constant?
- The value of the spring constant κ must depend on the degree of tissue displacement. In that regard, is it reasonable to use constant spring constant κ (7.6 N/m) even in the case that the

tissue displacement is non negligible?

- In line 134 of the Supplementary Material. is "r" different from "R"?

Reviewer #3 (Remarks to the Author):

This is a heroic effort in which pigs are used to test a hypothesis that lungs can respond to an acute bacterial challenge with enhanced fluid secretion, and that this response is defective in cystic fibrosis. This is an important question, with obvious therapeutic implications, but the ability to ascertain the answer to this problem has been previously limited by lack of appropriate animal models and techniques to measure secretion of small volumes in the lungs in intact animals. Here the authors have used and validated a sophisticated imaging technique that has enabled them to measure the height of the airway surface liquid in wild-type and CF pigs. In addition, tracheas from these animals were studied *in vitro*. The authors find that the pig lung *in vivo* responds acutely to an intra-tracheal bacterial challenge with enhanced fluid secretion that is absent in CF lungs. This is a significant observation. They further demonstrate the specificity for particular bacteria in eliciting this response, and demonstrate using *ex vivo* tracheas that a basal secretion is present in WT pigs that is CFTR dependent and apparently not under neuronal control, and which is deficient in CF pigs.

Although I have thought of some experiments that could extend these observations, I am loath to because the expense and work involved in the present study is considerable. I would however make a few points that the authors can consider to perhaps help with clarity in a revised manuscript.

1. The average ASL height in wild-type swine trachea is about 80 μm . Perhaps the authors could comment on this. That is, it was my impression that the ASL height should be about that of the length of the cilia to enable mucociliary clearance. Here the height is greater, or is it not?
2. The authors suggest (line 97) that the CF swine have "a considerably reduced basal level of ASL secretion (Fig 2d)." But I don't think that Fig 2d shows this. Do the authors mean that comparing the 30 min delta in ASL height, in the absence of bacteria, is less in CF, i.e. comparing the black bars in Fig 2c and 2f? If this is the case, they should make this clear. But in addition, because the height under basal conditions should reflect the balance of processes that include but not limited to secretion, is it accurate for the authors to refer to that difference as a difference in basal "secretion"?
3. I'm a bit confused by the labeling of columns in the various figures. Is this correct: In comparing columns labeled "a", they are not significantly different. Is this true even across panels (for example, can I compare column "a" in Fig 2c with columns labeled "a" in 2f?). If I see two comparisons but only one P value, then the P value is true for both those comparisons? If I see three labels, is this also true? For example, in Fig 3a, should I assume that PAK is different from Bac-free and also different from *delFliC* but not from FLA, and that Bac-free is different from *delFliC*? All with the same P value?
4. The implication from line 120 is that the airway cells are secreting cytokines in response to the bacteria-laden beads, and it is the submucosal glands that respond to those factors by secreting fluid. Is that what is meant? Have the authors tested this?
5. Fig 4a. I assume it's the isolated trachea? It is not clear if this is the *in vivo* or *in vitro* preparation, and what the before is and what the after is, in order to calculate the delta.
6. Is it significant that the CF secretion rate in Fig 4a is greater than that of WT tissue without

SMG or inhibited by Inh172? SMG have calcium-activated chloride channels,,,are they playing a role in the CF trachea here?

Reviewers' comments:

Reviewer #1 (Remarks to the Author):

The manuscript by Luan et al employ a novel in vivo imaging method (synchrotron-based x-ray imaging) to probe mechanisms of airway secretion in wild type and cystic fibrosis (CF) pigs. Using this model, they demonstrate that CFTR-dependent basal and toll-like receptor-mediated airway surface liquid (ASL) secretion occur in response to bacterial exposure (or bacterial components) and that this mechanism fails in CF pig trachea. These experiments attempt to fill in gaps in knowledge regarding how the CFTR-deficient airway differs from normal at homeostasis and when perturbed by daily bombardment of inhaled bacteria. This nicely complements and expands upon prior in vitro experiments conducted in human polarized ALI published by the UNC Chapel Hill group (Button et al, *Sci Signaling* 2013), and links that work to other aspects of the emerging model of impaired host defenses in the CF airway. However, the findings in this manuscript also contradict studies in the CF pig model by the Welsh group that demonstrate no abnormalities in ASL height in vivo (*Cell*. 2010 143:911-23). Nonetheless, other groups such as Rowe et al. show that ASL height in excised CF pig tracheal is lower than wild type using microoptical coherence tomography (*Am J Respir Crit Care Med*. 2014 190:421–432). Regardless of the CF ASL debate, the strength of science in this manuscript pertains to the induced secretory response by bacterial pathogens and the new in vivo method for assaying ASL height in vivo.

There are few studies in CF pigs by groups outside University of Iowa, so I do think this adds to the significance of this manuscript and also points out limitations of the gut-corrected pig model that are not evident in previous publications. Given the debate on ASL height in CF pigs, this study also favors the prevailing model in the field of airway dehydration in CF, which is useful.

Major concerns:

1. There is one major scientific variable that could significantly impact the outcome of experiments. Table S1, and the paragraph in which it is cited, reports the health problems with the piglets. Initial reports of the model suggest that meconium ileus typically leads to septic shock by 48 hrs. This present study reports that gut-correction does not completely rescuing these gut related health problems. Seven of the eleven CF animals had intestinal complication and thus were likely septic at the time of their health failure. I am not sure what happens to the activation of toll-like receptors on the basal lateral membrane of airway epithelia, but it could certainly down-regulate the receptor from the apical surface of the epithelia. So the question is, does sepsis interfere with airway secretion and/or bacterial responses that promote airway secretions. The only way I can think to rule this out is to make WT pigs septic and repeat the studies. If secretions are reduced, then much of what they might be studying is a septic response in the lungs. This would still be interesting, but a different conclusion of their in vivo data.

Response:

The evidence indicates that septicemia cannot account for our results for the following reasons

- a) ***The pathology analysis did not find evidence of septicemia:*** There is no evidence of septicemia (septic shock) on gross or microscopic examination in any of the animals studied. Septicemia in pigs would usually display petechial or ecchymotic hemorrhage in subcutaneous tissues, serosal surfaces of different organs, and renal parenchyma. We would also expect to observe pulmonary edema, and gastric fundic congestion. None of these gross anatomical lesions were observed in any animal. Septicemia would also cause microscopic lesions including hemorrhage and fibrin thrombi in different organs, especially in the stomach and lungs, as well as cause bacterial emboli in various organs. None of the above changes were observed by the board-certified pathologist Dr. Yanyun Huan who, based on gross and microscopic examinations, confidently concluded that septicemic shock was unlikely in any of the animals used in this study.
- b) ***CFTR^{-/-} animals that did not have gut problems display abnormal response to bacteria:*** CFTR^{-/-} animals with healthy guts (#1, 7, and 9) also displayed a reduced response to bacteria (Fig. A). In addition, two of those animals (# 1 and 9) were subjected to urinalysis and blood analysis (see reports attached below), which show that these animals have normal parameters, with a slight anemia, indicating that they did not suffer from septicemia.

Figure A: Response to *P. aeruginosa*-laden and bacteria-free beads by CFTR^{-/-} animals that did not develop gut problems (animals #1, 7 and 9). The results show that these animals that have not gut issues also fail to respond to bacteria, confirming that the gut abnormality displayed by some animals does not explain the failure to respond to bacteria by CFTR^{-/-} swine.

Thus, one must conclude that the reduced response to bacteria observed in CFTR^{-/-} swine cannot be explained by a potential septic condition.

2. Do they have evidence that dissecting away submucosal glands, which reduced basal secretions, does not damage the surface airway epithelium and its ability to secrete fluid?

Response:

Yes, we have evidence that dissecting away submucosal glands does not damage the surface airway epithelium.

In a previous publication, Luan, X. et al. (2014) *Proc. Natl. Acad. Sci. USA* 111, 12930-12935, we tested the effect of removing submucosal glands on the surface epithelia. As stated in the publication (page 12934 and 9th paragraph): “We tested the condition of the surface epithelium after dissection of submucosal glands by measuring short circuit currents in an Ussing chamber. The experiments showed that preparations subjected to the dissection procedure (removal of the cartilage) but with submucosal glands intact had a short circuit current (I_{sc}) of 24 $\mu\text{A}/\text{cm}^2$. Similarly, preparations where we dissected the glands out had a I_{sc} of 20 $\mu\text{A}/\text{cm}^2$. Addition of amiloride (10^{-4} M apical side) reduced the I_{sc} to 12 and 9 $\mu\text{A}/\text{cm}^2$ for preparations with and without glands, respectively. Stimulation with forskolin (10^{-5} M apical side) stimulated I_{sc} to 20 and 17 $\mu\text{A}/\text{cm}^2$ for preparations with and without glands, respectively. These results showed that the surface epithelium was not affected by the dissection procedure.”

Minor concerns:

1. Figure 3e is called 3f in the legend.

Response: Error fixed

In the text describing panels d and e, they don't explicitly state that IL6 and IL8 stimulate secretions in WT, they just state it does not do so in CF. It would be good to reword this to point out both aspects.

Response:

We have changed the figure legend to read:

“Secretion assay data showed that **d**) IL-6 (50 ng/mL, n = 16 preparations from 7 CFTR -/- animals and n = 9 preparations from 4 wild-type animals, $p < 0.0001$, $F(3, 60) = 12.79$, ANOVA and Tukey's multiple comparison test) and **e**) IL-8 (50 ng/mL, n = 15 for CFTR -/- airway tissue from 7 pigs and n = 10 for wild type tissue from 4 pigs, $p < 0.0001$, $F(3, 56) = 13.09$, ANOVA and Tukey's multiple comparison test) both trigger submucosal gland secretion in wild-type airway, but fail to stimulated ASL secretion in CF airway.”

2. In generally the Figure legends are pretty wordy since they repeat all the statistical tests for each

panel. It might be easier to read if they grouped the statistical test at the end of the legend for various levels of significance and if the N varies, they could write that next to the genotype below the x-axis.

Response:

The journal has very specific “Reporting requirements for life sciences research” (http://www.nature.com/article-assets/npg/ncomms/authors/ncomms_lifesciences_checklist.pdf) that mandate this information be stated in the figure legend as well. Therefore, the repetitiveness is rather unavoidable.

3. It might be useful to point out that piglets were delivered by c-section to avoid bacterial exposure to the lung during birth. Presumably this is why they did that.

Response:

After consultation with a number of veterinarians who have had considerable experience working with CFTR^{-/-} swine, we decided to perform a cesarean section to reduce the stress of natural birth on the piglets and also to reduce bacterial exposure in the birth canal. We have made this point more clear in Supplementary data 1, line 7, where we have stated: “To reduce exposure to bacteria in the birth canal and to minimize stress of natural birth on the piglets, the delivery was done through a caesarean section performed by a specialized veterinary surgeon team to ensure minimal exposure of the piglets to anaesthetics.”

4. Since this is one of the only published studies in the gut-corrected pigs outside the Iowa group and this present study evaluated gut expression of the CFTR transgene, if they have the tissue it would be useful to include CFTR mRNA expression studies in the airways of CF pigs to confirm it is lower than WT and that the transgene is not ectopically expressed.

Response:

The expression of the CFTR transgene in the airway of CFTR^{-/-} gut-corrected swine has already been tested by Stoltz et al. (2013). Using quantitative qPCR (Stoltz et al 2013, Supplemental Figure 2A) showed that CFTR expression level in the airway of the CFTR^{-/-} gut-corrected swine is very low, and comparable with that of CFTR^{-/-} non-gut-corrected swine.

In addition, the gut-corrected CFTR^{-/-} swine also display the anatomical and physiological abnormalities expected from a cystic fibrosis airway. Stoltz et al 2013 show that gut corrected CFTR^{-/-} swine display pancreatic destruction, liver disease, reduced weight gain, and developed sinus and lung disease. In addition, the airways of these animals display electrophysiological abnormalities that are similar to those observed in CFTR^{-/-} swine and CF patients. Similarly, our submucosal gland secretion assay results are consistent with those expected from CF airway.

Further, conducting a budget- and time-consuming qPCR experiment to determine CFTR mRNA expression levels in airways, will add little to the article.

Finally, our stored tissue samples have deteriorated over time, and recent tests of RNA quality were unsatisfactory and it would be difficult to successfully perform accurate qPCR analyses with them.

5. They excluded one CF piglet that was particularly affected with respiratory failure from birth (suppl material, line 14). However, they also state that they did not exclude any data (main body, line 288). The statements should be consistent.

Response:

We clarify this inconsistency in line 299 in the main body of the manuscript: “Animal #11 was not included in the study because it never recovered from the stress of delivery. We did not exclude any data collected from any of the animals studied from our analysis.”

Reviewer #2 (Remarks to the Author):

This study addresses important questions, whether the inhalation of bacteria triggers airway surface liquid (ASL) secretion and whether this process is abnormal in cystic fibrosis (CF). The authors developed a new technique to approach these issues. However, I have several concerns and question regarding the manuscript:

- The authors noted “The main challenge in testing this hypothesis is the lack of an imaging method.....”, it is not clear why a highly specialized synchrotron-based x-ray imaging technique is necessary to measure the ASL layer, instead of a more common, commercially available, portable x-ray micro CT system applicable for live animals, which has sufficient spatial resolution. The authors need to clarify this.

Response:

It is hard to answer this question without clarifying which specific portable x-ray micro CT device the reviewer has in mind. We are not aware of any portable x-ray micro CT system that would allow us to perform phase contrast imaging and K-edge subtraction techniques *in vivo* on a 5 kg animal, and is capable of sufficient resolution to observe the ASL layer. As far as we know, only the Canadian Light Source BMIT beamline can collect this data.

To our knowledge, commercially available x-ray micro CT systems lack the in-line phase contrast capability that we use to see the air/ASL interface. Also, we use projection images in our research while CT will use several hundred images and thus the overall exposure to radiation to the animal would be very much higher and possibly harmful to the animal.

In addition, we rely on the tunability of synchrotron radiation to use the K-edge subtraction method to determine which beads contain iodine or barium contrast elements. This element-specific sorting method is used to distinguish bacteria-laden and bacterial-free beads which constitute an internal control in the experiments. Commercially available portable x-ray micro CT devices are unable to perform K-edge subtraction due to the broad banded nature of conventional x-ray tube used in such systems. Commercially available portable x-ray micro CT devices systems may be able to detect higher-Z elements of iodine or barium, but will not be able to distinguish the elements from each other.

- The authors stated that “the air-ASL interface can be detected only at the ventral (top) and at the dorsal (bottom) of the trachea....”, therefore, the animal (for in vivo imaging) or the harvested trachea (for ex vivo) was rotated to ensure that beads were positioned exactly at the top and at the bottom of the trachea using a computer-controlled motorized stage. The placement of the beads to precise positions (exactly ventral and dorsal) must be critical for the experiment and needs to be done in micron precision. Please provide more detailed description of this procedure.

Response:

We have included this information in line 245:

“Since we had no prior knowledge of the exact position of the agarose beads, we rotated the sample (i.e. the anesthetized swine or isolated trachea) along its longitudinal axis 1 degree at a time and recorded an image. We repeated this process for 23 degrees. The ASL height was measured using those beads that reached the top or bottom of the trachea, at which point the Air/ASL interface becomes visible.”

Contrary to the statement by the reviewer we do not need a micron scale precision for our measurement for the following two reasons:

- a) The resolution of our measurements is limited by the pixel size of our detector (8.75 x 8.75 μm). Thus, only positioning errors in the range of tens of microns could be detected in our experiments.
- b) All of our conclusions are based **not on the absolute value** of ASL but rather on the change in ASL between 5 min and 35 min measurements. Thus, any error in our measurement in ALS height due to positioning of the beads would be cancelled by the subtraction of ASL at 35 minus 5 minutes.

- Is placing the plane of the x-ray exactly at the middle of a bead (Figure S3) crucial for applying the mathematical model the authors use? If so, please provide more detailed description of how this is done (the experimental procedure).

Response:

No, the plane of incidence of the x-ray has no bearing on the mathematical model. We use the model to provide a theoretical background for the ASL/bead interaction to complement the experimental studies reported in the supplementary data. The main conclusion of the mathematical model is that the surface tension of the ASL is strong enough to retain the beads against the tissue, but it does not cause serious deformation of the airway surface epithelia. The x-rays are not part of this model.

- Since it is crucial for the measurement that the agarose beads are immobilized on the surface of the epithelium due to the surface tension of the ASL, a video demonstration as a supplementary material would be nice. Although the authors state this on lines 86-90 in the text, it would be easier for the reader to see it as a video.

Response:

We cannot produce a 30 min long exposure of the preparation to x-rays since it would exceed the radiation dose allowed, which is inhumane, and it may damage the tissue, causing artifacts. In the figure B (below) we show representative images of the same preparation taken 5 min and 35 min. The images demonstrate that the beads are not susceptible to mucociliary clearance and do not move towards the mouth (i.e. to the right of the image) during an experiment. The beads are immobile.

Figure B: Agarose beads are immobilized in the trachea of swine during the experimental period. The agarose beads (yellow and red arrows) do not move over the 30 min time period. The lead tape is pasted on the skin of the pig to provide a fixed point in the image and be used as an indicator of the position of the beads.

- The authors use the average value of 32 mN/m of airway fluid surface tension (γ) in their calculation, is this value reasonable in the case of CF?

Response:

The difference in surface tension between CF and wild-type ASL would have no effect on our measurements.

The surface tension of ASL in CF is 81.1 mN/m (Albers et al., 1996 *J. Appl. Physiol.* 81(6): 2690–2695). The higher surface tension will increase both the adhesion of the beads to the tissue and the tissue displacement. However, there would be no effect on our measurements:

- a) *Increased adhesion to the tissue:* the larger surface tension would increase the adhesion force exerted on beads by the ASL layer. This would make it more likely that the beads would be immobilized against the airway surface.
- b) *Increased tissue displacement has negligible effect on our conclusions.* Since we based all our conclusion on the change in ASL layer height over 30 min, and the tissue displacement for each bead would be the same at 5 and 35 min. Thus, any difference in ASL measurement caused by tissue displacement between CF and wild-type would be cancelled by the subtraction of ASL height at 35 minus 5 minutes.

Moreover, the effect of higher surface tension on tissue displacement goes against our hypothesis since it would cause an overestimation of ASL in CF. Thus, any potential artifacts of increased tissue displacement would go against our conclusion that CF tissue produce less ASL than wild-type.

Finally, the increase in tissue displacement would have marginal effect on our measurements. An agarose bead of average size (~700 μ m diameter) causes a tissue displacement of ~8 μ m. The same bead in CF tissue would cause a tissue displacement of ~20 μ m. However, since the resolution of our experiment is limited by the detector pixel size (8.75x8.75 μ m) both the CF and wild-type displacement are at or near the resolution limit of our experiment. In other words, we would not be able to see the difference between CF and wild-type displacement on our ASL measurement.

- Does the viscosity of ASL change in patients with CF? If it does, can the surface tension of ASL be considered constant?

Response:

There are reports that the viscosity of the ASL is different in CF and that it may play a role in CF pathology (*Am J Respir Crit Care Med.* 2014 190:421–432). The viscosity of the ASL layer is directly related to the surface tension which is greater in CF ASL, and the increased surface tension is discussed in the previous comment. Thus, changes in viscosity influence the adhesion force of the beads against the tissue in the same manner that increased surface tension does.

- The value of the spring constant κ must depend on the degree of tissue displacement. In that regard, is it reasonable to use constant spring constant κ (7.6 N/m) even in the case that the tissue displacement is non negligible?

Response:

For the spring constant to change there needs to be an inelastic permanent deformation of the tissue. Over the negligible displacement distances suffered by the airway surface epithelia this is very unlikely. Thus, it is safe to assume that the spring constant does not change over the range of tissue displacements in our experiment.

• In line 134 of the Supplementary Material. is “r” different from “R”?

Response:

They are the same. We have fixed this error.

Reviewer #3 (Remarks to the Author):

This is a heroic effort in which pigs are used to test a hypothesis that lungs can respond to an acute bacterial challenge with enhanced fluid secretion, and that this response is defective in cystic fibrosis. This is an important question, with obvious therapeutic implications, but the ability to ascertain the answer to this problem has been previously limited by lack of appropriate animal models and techniques to measure secretion of small volumes in the lungs in intact animals. Here the authors have used and validated a sophisticated imaging technique that has enabled them to measure the height of the airway surface liquid in wild-type and CF pigs. In addition, tracheas from these animals were studied in vitro. The authors find that the pig lung in vivo responds acutely to an intra-tracheal bacterial challenge with enhanced fluid secretion that is absent in CF lungs. This is a significant observation. They further demonstrate the specificity for particular bacteria in eliciting this response, and demonstrate using ex vivo tracheas that a basal secretion is present in WT pigs that is CFTR dependent and apparently not under neuronal control, and which is deficient in CF pigs.

Although I have thought of some experiments that could extend these observations, I am loath to because the expense and work involved in the present study is considerable. I would however make a few points that the authors can consider to perhaps help with clarity in a revised manuscript.

1. The average ASL height in wild-type swine trachea is about 80 μm . Perhaps the authors could comment on this. That is, it was my impression that the ASL height should be about that of the length of the cilia to enable mucociliary clearance. Here the height is greater, or is it not?

Response:

The airway surface liquid (ASL) layer is composed of two different phases, a mucus layer and a periciliary liquid layer (PCL). The height of the PCL should be about that of the length of the cilia. However, our measurements include both the PCL and the mucus layer. The height measured in our experiments at 5 min is in agreement with previous reports of ASL height (Worthington and Tarran, 2011, *Methods Mol Biol.* 742:77–92).

2. The authors suggest (line 97) that the CF swine have “a considerably reduced basal level of ASL secretion (Fig 2d).” But I don’t think that Fig 2d shows this. Do the authors mean that comparing the 30 min delta in ASL height, in the absence of bacteria, is less in CF, i.e. comparing the black bars in Fig 2c and 2f? If this is the case, they should make this clear.

Response:

We agree with the reviewer that this section is confusing. To clarify this issue we have deleted the statement about basal secretion from the section highlighted by the reviewer (line 97) and included made it more clear in line 125.

But in addition, because the height under basal conditions should reflect the balance of processes that include but not limited to secretion, is it accurate for the authors to refer to that difference as a difference in basal “secretion”?

Response:

Yes, the term secretion is appropriate. The basal ASL secretion, i.e. ASL secretion without bacterial stimulation, can be blocked by blocking ion transport by epithelial cells or removing airway submucosal glands (Fig. 4b and c). These results indicate that this basal ASL is secreted by the airway epithelia.

3. I’m a bit confused by the labeling of columns in the various figures. Is this correct: In comparing columns labeled “a”, they are not significantly different. Is this true even across panels (for example, can I compare column “a” in Fig 2c with columns labeled “a” in 2f?).

Response:

No, the statistical analysis is valid only within each panel. For example, the data in Fig. 2c has been analyzed using statistics that only apply to Fig. 2c and cannot be extrapolated to Fig. 2f or any other panel or figure. We have made this clear in the figure legends.

If I see two comparisons but only one P value, then the P value is true for both those comparisons? If I see three labels, is this also true? For example, in Fig 3a, should i assume that PAK is different from Bac-free and also different from DelfliC but not from FLA, and that Bac-free is different from delFliC? All with the same P value?

Response:

That is correct. Within each panel, columns labeled with different letters are statistically different. Whereas those labeled with the same letter are not.

In Fig. 3 all tests are ANOVA with Tukey's multiple comparison test. The p value reported corresponds to the ANOVA, we do not report the p value of the multiple comparisons other than "p < 0.05 was considered significant" as stated in line 299.

4. The implication from line 120 is that the airway cells are secreting cytokines in response to the bacteria-laden beads, and it is the submucosal glands that respond to those factors by secreting fluid. Is that what is meant? Have the authors tested this?

Response:

Yes, that is what is meant. It is a working hypothesis based on the observations that:

- a) The response to *P. aeruginosa* is heavily dependent on expression of flagellin, a ligand for TLR5
- b) The airway respond to LPS, a ligand for TLR 4.
- c) Cytokines normally released in response to activation of TLRs triggers ASL secretion by submucosal glands (Fig. 3; Luan et al., 2014, *Proc. Natl. Acad. Sci. USA* 111, 12930-12935)

We have not yet directly tested this hypothesis in CFTR^{-/-} swine.

5. Fig 4a. I assume it's the isolated trachea? It is not clear if this is the in vivo or in vitro preparation, and what the before is and what the after is, in order to calculate the delta.

Response:

No, Fig. 4a is from live wild-type and CFTR^{-/-} animals. We have clarified this by adding the label "in vivo" to that panel.

In that panel we are comparing the bacteria-free beads measurements in wild-type and CFTR^{-/-} swine that we reported in Fig. 2.

6. Is it significant that the CF secretion rate in Fig 4a is greater than that of WT tissue without SMG or inhibited by Inh172? SMG have calcium-activated chloride channels,,are they playing a role in the CF trachea here?

Response:

The data presented in Fig. 4a is from living animals while the WT without SMG or inhibited by CFTRinh172 was produced from isolated trachea. In general, in vivo preparations produce more secretion than ex vivo preparations. Thus, it is difficult to compare in vivo with ex vivo data.

The possible role of Ca^{2+} activated Cl^- channels in CF tracheas was not directly tested. However, in the experiments presented in Fig. 3d and e, as well as Joo et al., 2010 (*J Clin Invest* 120, 3161-3166), we stimulated the preparations with carbachol, a cholinergic agonist, as a positive control to demonstrate the viability of the preparations. The data show that CFTR^{-/-} glands respond to carbachol, and this response has been previously shown to involve Ca^{2+} activated Cl^- channels.

Test Report

External Id: Pig 1 Species: Porcine Breed: Porcine (P) Sex: Unknown Age: 7.00 Day(s)	Incident/Project Number: 1-415429 Submitted: 05-Mar-2016 Collected: NA Veterinarian: Ianowski, Juan Owner: Ianowski, Juan Copy To: Copy To: Samples Submitted: EDTA x 1, Serum x 1, Slide x 3
RS-Ianowski, Juan Health Sciences Building Room 2D30.4 -107 Wiggins Road Saskatoon, SK S7N 5E5 Phone: (306) 966-2542 Fax: (306) 966-4298	

History: Research

Contents: Chemistry Analyzer / Large Animal Standard (Chemistry Panel) (FINAL) x 1, Complete Blood Count (FINAL) x 1

Chemistry Analyzer - Final

Collected: NA	Received: 05-Mar-2016	Tested: 05-Mar-2016	Completed: 05-Mar-2016
---------------	-----------------------	---------------------	------------------------

Chemistry Analyzer / Large Animal Standard (Chemistry Panel)

ID	Sample	Test	Result	Units	Reference	Flag	Comment
Pig 1	Serum	Sodium	143	mmol/L			
Pig 1	Serum	Potassium	4.6	mmol/L			
Pig 1	Serum	Chloride	99	mmol/L			
Pig 1	Serum	Bicarbonate	36	mmol/L			
Pig 1	Serum	Anion Gap	13	mmol/L			
Pig 1	Serum	Calcium	2.54	mmol/L			
Pig 1	Serum	Phosphorus	2.59	mmol/L			
Pig 1	Serum	Magnesium	1.34	mmol/L			
Pig 1	Serum	Urea	9.1	mmol/L			
Pig 1	Serum	Creatinine	31	µmol/L			
Pig 1	Serum	Glucose	3.3	mmol/L			
Pig 1	Serum	Total Bilirubin	1.2	µmol/L			
Pig 1	Serum	Direct Bilirubin	0.4	µmol/L			
Pig 1	Serum	Indirect Bilirubin	0.8	µmol/L			
Pig 1	Serum	GGT	97	U/L			
Pig 1	Serum	GLDH	0	U/L			
Pig 1	Serum	AST	13	U/L			
Pig 1	Serum	CK	161	U/L			
Pig 1	Serum	Total Protein	35	g/L			
Pig 1	Serum	Albumin	17	g/L			
Pig 1	Serum	Globulin	18	g/L			
Pig 1	Serum	A:G Ratio	0.94				

Sample	Lipemia	Hemolysis	Yellow
Serum	None	None	None

Authorizing Signature: (name deleted)

Date: 07-Mar-2016

Area Supervisor Clinical Pathology

Complete Blood Count Pig 1 - Final

Collected: NA Received: 05-Mar-2016 Tested: 07-Mar-2016 Completed: 07-Mar-2016

Leukocytes	Result		Flag	Ref. Int. x10 ⁹ /L
WBC				11.0 - 22.0
Corrected WBC	10.8		L	11.0 - 22.0
NRBC /100 WBCs	23			
Differentials	Rel%	Abs	Flag	Ref. Int. x10 ⁹ /L
Segs	46	4.968		3.080 - 10.400
Bands	7	0.756		0.000 - 0.880
Metamyelocytes				
Myelocytes				
Toxic Change	Slight			
Eosinophils	1	0.108		0.055 - 2.420
Basophils				0.000 - 0.440
Lymphocytes	37	3.996	L	4.290 - 13.600
Monocytes	9	0.972		0.220 - 2.200
Atypical				
Other				

Few reactive lymphs / monos

Platelets	Value	Flag	Ref. Int. x10 ⁹ /L
Platelet Clumped	Yes		
Estimate (slide)	Increased		
Morph (slide)	Normal		
Auto Count (min.)	722		250 - 850

Platelets appear slightly increased

Erythrocytes	Result	Flag	Reference	Units
RBC	3.13	L	5.00 - 8.00	x10 ¹² /L
HGB	68	L	100 - 160	g/L
HCT	0.246	L	0.320 - 0.500	L/L
MCV	78.8	H	50.0 - 68.0	fL
MCH	21.6		16.6 - 22.0	pg
MCHC	275	L	300 - 340	g/L
RDW	22.0			%
Retics	15.0			%
RPI				

RBC Morphology
Anisocytosis 3+, Macrocytosis 2+, Microcytosis 1+, Nuclear Remnants Few, Echinocyte 1 3+, Polychromasia 3+, Unclassified 2+

Plasma Protein by Refractometry	Result	Flag	Ref. Int. g/L
Total Protein	41		
Fibrinogen	1		
Total Protein: Fib Ratio	41:1		

Substances that artifactually increase total protein by refractometry include urea, glucose, cholesterol, lipoproteins and excess anticoagulant.

See chemistry panel for appearance

Authorizing Signature: (name deleted)

Date: 07-Mar-2016

, Area Supervisor Clinical Pathology

Test Report

External Id: #1 Species: Porcine Breed: Porcine (P) Sex: Unknown Age: 7.00 Day(s)	Incident/Project Number: 1-415429-1120-account-8000-15535 Submitted: 04-Mar-2016 Collected: NA Veterinarian: Ianowski, Juan Owner:
RS-Ianowski, Juan Health Sciences Building Room 2D30.4 -107 Wiggins Road Saskatoon, SK S7N 5E5 Phone: (306) 966-2542 Fax: (306) 966-4298	Copy To: Copy To: Samples Submitted: Urine x 1

History:
 This animal is CFTR-/- for research. Please do urinalysis.

Contents: Complete Urinalysis (FINAL) x 1

Urinalysis #1 - Final

Collection Method: Urine (plastic tube) - Unknown

Collected: NA	Received: 04-Mar-2016	Tested: 05-Mar-2016	Completed: 05-Mar-2016
---------------	-----------------------	---------------------	------------------------

PHYSICAL	Result
Clarity/Colour	clear colorless
Specific Gravity	1.004
REAGENT STRIP	Result
pH	9.0
Protein	negative
Glucose	trace
Ketones	negative
Bilirubin	negative
Blood	negative

SEDIMENT	Result
WBC /hpf	
RBC /hpf	
Epithelial Cells/hpf	
Crystals/hpf	scant struvite
Casts/lpf	
Bacteria Rods/hpf	scant
Bacteria Cocci/hpf	scant
Fat/hpf	
Other	scant debris

Authorizing Signature: (name deleted)

Date: 07-Mar-2016

Area Supervisor Clinical Pathology

Test Report

External Id: 9 Species: Porcine Breed: Porcine (P) Sex: Unknown Age: 7.00 Day(s)	Incident/Project Number: 1-415429-1120-8000-15535 Submitted: 04-Mar-2016 Collected: NA Veterinarian: Ianowski, Juan Owner: Ianowski, Juan #9 Copy To: Copy To: Samples Submitted: EDTA x 1, Serum x 1, Urine x 1
RS-Ianowski, Juan Health Sciences Building Room 2D30.4 -107 Wiggins Road Saskatoon, SK S7N 5E5	
Phone: (306) 966-2542 Fax: (306) 966-4298	

History:
 CFTR pig for research. UA, blood.

Contents: Culture(s) (FINAL) x 1, Chemistry Analyzer / Large Animal Standard (Chemistry Panel) (FINAL) x 1, Complete Blood Count (FINAL) x 1, Complete Urinalysis (FINAL) x 1

Bacteriology - Final

Comment:

There is no evidence of UTI.

Collected: NA	Received: 04-Mar-2016	Tested: 05-Mar-2016	Completed: 07-Mar-2016
---------------	-----------------------	---------------------	------------------------

Culture(s)

ID	Sample	Target	Type	Result	Units	Comment
9	Urine	Negative culture				

Authorizing Signature: (name deleted)

Date: 07-Mar-2016

, Veterinary Microbiologist

Chemistry Analyzer - Final

Collected: NA	Received: 04-Mar-2016	Tested: 05-Mar-2016	Completed: 05-Mar-2016
---------------	-----------------------	---------------------	------------------------

Chemistry Analyzer / Large Animal Standard (Chemistry Panel)

ID	Sample	Test	Result	Units	Reference	Flag	Comment
9	Serum	Sodium	146	mmol/L			
9	Serum	Potassium	4.5	mmol/L			
9	Serum	Chloride	101	mmol/L			
9	Serum	Bicarbonate	38	mmol/L			
9	Serum	Anion Gap	12	mmol/L			
9	Serum	Calcium	2.58	mmol/L			
9	Serum	Phosphorus	2.30	mmol/L			
9	Serum	Magnesium	1.02	mmol/L			
9	Serum	Urea	1.9	mmol/L			
9	Serum	Creatinine	21	µmol/L			

Collected: NA	Received: 04-Mar-2016	Tested: 05-Mar-2016	Completed: 05-Mar-2016
---------------	-----------------------	---------------------	------------------------

Chemistry Analyzer / Large Animal Standard (Chemistry Panel)

ID	Sample	Test	Result	Units	Reference	Flag	Comment
9	Serum	Glucose	5.1	mmol/L			
9	Serum	Total Bilirubin	1.7	µmol/L			
9	Serum	Direct Bilirubin	0.4	µmol/L			
9	Serum	Indirect Bilirubin	1.3	µmol/L			
9	Serum	GGT	91	U/L			
9	Serum	GLDH	0	U/L			
9	Serum	AST	19	U/L			
9	Serum	CK	102	U/L			
9	Serum	Total Protein	25	g/L			
9	Serum	Albumin	10	g/L			
9	Serum	Globulin	15	g/L			
9	Serum	A:G Ratio	0.67				

Sample	Lipemia	Hemolysis	Yellow
Serum	None	None	None

Authorizing Signature: (name deleted)

Date: 07-Mar-2016

, Area Supervisor Clinical Pathology

Complete Blood Count 9 - Final

Collected: NA	Received: 04-Mar-2016	Tested: 05-Mar-2016	Completed: 05-Mar-2016
---------------	-----------------------	---------------------	------------------------

Leukocytes	Result	Flag	Ref. Int. x10 ⁹ /L
WBC			11.0 - 22.0
Corrected WBC	11.5		11.0 - 22.0
NRBC /100 WBCs	6		
Differentials	Rel%	Abs	Flag
Segs	66	7.590	
Bands			
Metamyelocytes			
Myelocytes			
Toxic Change			
Eosinophils	1	0.115	
Basophils			
Lymphocytes	25	2.875	L
Monocytes	8	0.920	
Atypical			
Other			

Few hypersegmented neutrophils, reactive lymphs

Platelets	Value	Flag	Ref. Int. x10 ⁹ /L
Platelet Clumped	Yes		
Estimate (slide)	Increased		
Morph (slide)	Normal		
Auto Count (min.)	743		250 - 850

Platelets appear moderately increased

Authorizing Signature: (name deleted)

Date: 09-Mar-2016

, Area Supervisor Clinical Pathology

Erythrocytes	Result	Flag	Reference	Units
RBC	3.35	L	5.00 - 8.00	x10 ¹² /L
HGB	68	L	100 - 160	g/L
HCT	0.244	L	0.320 - 0.500	L/L
MCV	72.8	H	50.0 - 68.0	fL
MCH	20.3		16.6 - 22.0	pg
MCHC	279	L	300 - 340	g/L
RDW	20.1			%
Retics	13.2			%
RPI				

RBC Morphology
Anisocytosis 2+, Macrocytosis 2+, Microcytosis 1+, Nuclear Remnants Few, Echinocyte 1 2+, Polychromasia 2+, Unclassified 1+

Plasma Protein by Refractometry	Result	Flag	Ref. Int. g/L
Total Protein	38		
Fibrinogen	2		
Total Protein: Fib Ratio	19:1		

Substances that artifactually increase total protein by refractometry include urea, glucose, cholesterol, lipoproteins and excess anticoagulant.

Urinalysis 9 - Final

Comment:

~2.0 mls rec'd

Collection Method: Urine - Unknown

Collected: NA	Received: 04-Mar-2016	Tested: 05-Mar-2016	Completed: 05-Mar-2016
---------------	-----------------------	---------------------	------------------------

PHYSICAL	Result
Clarity/Colour	clear colorless
Specific Gravity	1.004
REAGENT STRIP	Result
pH	9.0
Protein	negative
Glucose	normal
Ketones	negative
Bilirubin	negative
Blood	3+

SEDIMENT	Result
WBC /hpf	
RBC /hpf	0-2
Epithelial Cells/hpf	
Crystals/hpf	scant amorphous scant struvite
Casts/lpf	
Bacteria Rods/hpf	
Bacteria Cocci/hpf	
Fat/hpf	
Other	scant debris

Authorizing Signature: (name deleted)

Date: 07-Mar-2016

, Area Supervisor Clinical Pathology

Test Report

External Id: #9 Species: Porcine Breed: Porcine (P) Sex: Unknown Age: 7.00 Day(s)	Incident/Project Number: 1-415429-1120-account-8000-15535 Submitted: 04-Mar-2016 Collected: NA Veterinarian: Ianowski, Juan Owner:
RS-Ianowski, Juan Health Sciences Building Room 2D30.4 -107 Wiggins Road Saskatoon, SK S7N 5E5 Phone: (306) 966-2542 Fax: (306) 966-4298	Copy To: Copy To: Samples Submitted: Urine x 1

History:
 This animal is CFTR-/- for research. Please do urinalysis.

Contents: Complete Urinalysis (FINAL) x 1

Urinalysis #9 - Final

Collection Method: Urine (plastic tube) - Unknown

Collected: NA	Received: 04-Mar-2016	Tested: 05-Mar-2016	Completed: 05-Mar-2016
---------------	-----------------------	---------------------	------------------------

PHYSICAL	Result
Clarity/Colour	clear colorless
Specific Gravity	1.005
REAGENT STRIP	Result
pH	9.0
Protein	negative
Glucose	3+
Ketones	negative
Bilirubin	negative
Blood	negative

SEDIMENT	Result
WBC /hpf	
RBC /hpf	
Epithelial Cells/hpf	
Crystals/hpf	scant struvite
Casts/lpf	
Bacteria Rods/hpf	scant
Bacteria Cocci/hpf	scant
Fat/hpf	
Other	scant debris

Authorizing Signature: (name deleted)

Date: 07-Mar-2016

, Area Supervisor Clinical Pathology

REVIEWERS' COMMENTS:

Reviewer #1 (Remarks to the Author):

I am satisfied with the author's responses, but it would have been nice if they added something to the text regarding the clinical status and absence of a septic response in animals with meconium ileus. I really liked that panel they showed in the rebuttal Figure A, showing no difference between liquid layer responses in CFTR^{-/-} animals with and without meconium ileus. It would have been nice for the future readers of this work hear about that result as data not shown. That said, the tone of their responses to all the reviewers seems pretty obvious that this model is so much work, they want to be done with it.

Reviewer #2 (Remarks to the Author):

The paper has been improved. I have no further comments.

Reviewer #3 (Remarks to the Author):

The authors have addressed all of my concerns.

Response to the reviewer's comments

Reviewer #1:

“I am satisfied with the author's responses, but it would have been nice if they added something to the text regarding the clinical status and absence of a septic response in animals with meconium ileus. I really liked that panel they showed in the rebuttal Figure A, showing no difference between liquid layer responses in CFTR^{-/-} animals with and without meconium ileus. It would have been nice for the future readers of this work hear about that result as data not shown. That said, the tone of their responses to all the reviewers seems pretty obvious that this model is so much work, they want to be done with it.”

RESPONSE: We have added in page 10 line 223:

“The CFTR^{-/-} animals with healthy guts (#1, 7, and 9) displayed the same abnormal response to inhaled bacteria phenotype as those with gut problems (data not shown).”

And in page 11 line 232:

“There was no evidence of septicemia (septic shock) on gross or microscopic examination in any of the animals studied. Septicemia in pigs would usually display petechial or ecchymotic hemorrhage in subcutaneous tissues, serosal surfaces of different organs, and renal parenchyma. We would also expect to observe pulmonary edema, and gastric fundic congestion. None of these gross anatomical lesions were observed in any of the animals. Septicemia would also cause microscopic lesions including hemorrhage and fibrin thrombi in different organs, especially in the stomach and lungs, as well as cause bacterial emboli in various organs. None of the above changes were observed, thus, we concluded that septicemic shock was unlikely in any of the animals used in this study.”